# Deciphering Molecular Mechanisms and Intervening in Physiological and Pathophysiological Processes of Ca^2+^ Signaling Mechanisms Using Optogenetic Tools

**DOI:** 10.3390/cells10123340

**Published:** 2021-11-28

**Authors:** Lena Maltan, Hadil Najjar, Adéla Tiffner, Isabella Derler

**Affiliations:** Institute of Biophysics, JKU Life Science Center, Johannes Kepler University Linz, A-4020 Linz, Austria; lena.maltan@jku.at (L.M.); hadil.najjar@jku.at (H.N.); adela.tiffner@jku.at (A.T.)

**Keywords:** ion channels, calcium signaling, optogenetics, opsins, CRAC channel, light-sensitive Ca^2+^ permeable channels

## Abstract

Calcium ion channels are involved in numerous biological functions such as lymphocyte activation, muscle contraction, neurotransmission, excitation, hormone secretion, gene expression, cell migration, memory, and aging. Therefore, their dysfunction can lead to a wide range of cellular abnormalities and, subsequently, to diseases. To date various conventional techniques have provided valuable insights into the roles of Ca^2+^ signaling. However, their limited spatiotemporal resolution and lack of reversibility pose significant obstacles in the detailed understanding of the structure–function relationship of ion channels. These drawbacks could be partially overcome by the use of optogenetics, which allows for the remote and well-defined manipulation of Ca^2+^-signaling. Here, we review the various optogenetic tools that have been used to achieve precise control over different Ca^2+^-permeable ion channels and receptors and associated downstream signaling cascades. We highlight the achievements of optogenetics as well as the still-open questions regarding the resolution of ion channel working mechanisms. In addition, we summarize the successes of optogenetics in manipulating many Ca^2+^-dependent biological processes both in vitro and in vivo. In summary, optogenetics has significantly advanced our understanding of Ca^2+^ signaling proteins and the used tools provide an essential basis for potential future therapeutic application.

## 1. Introduction

### 1.1. Ca^2+^—The Versatile Signaling Protein

Calcium (Ca^2+^) ions are indispensable messengers in the human body and guarantee the function of the immune system, muscle contraction, and neuronal signaling. Those are established by a concerted interplay of millions of single cells. At the level of a single cell, these processes are finely tuned by a complex and sophisticated interplay of an immense variety of proteins that can sense or transport Ca^2+^ [1,2,3,4,5,6]. When the cell is at rest, the cytosolic Ca^2+^ concentrations are very low, not exceeding a threshold of 0.1µM. Activation of the cell by various stimuli, such as changes in the membrane potential or stimulation of receptors in the membrane, can induce an increase in Ca^2+^ levels across Ca^2+^ channels that respond to the particular stimulus. Overall, Ca^2+^ signals are translated into diverse biological output signals, ranging from short-term events, such as secretion or contraction, to long-term processes, such as gene transcription or proliferation [1,7,8]. Their versatility is ensured by diverse patterns of Ca^2+^ signals, including sustained and oscillatory Ca^2+^ signals, which are controlled by various proteins within the cell [9]. A single defect in a Ca^2+^ signaling protein or ion pore can lead to abnormal Ca^2+^ levels in the cell, which can be responsible for a variety of diseases, including severe immune deficiency and muscle or neurological disorders [10,11,12,13]. Both the physiological and pathophysiological roles of Ca^2+^ signaling proteins highlight their clinical relevance. Although there are cases where Ca^2+^ pharmacology turned out to be successful in therapy [14,15], Ca^2+^ ion channels are unfortunately still technically difficult in regard to target-specific drug discovery [16,17,18]. A detailed understanding of the structure–function relationship of essential Ca^2+^ signaling proteins and associated downstream signaling events is fundamental for the development of well-directed therapeutic approaches. Given the important role of Ca^2+^ ion channels in health and disease, the integration of optogenetics in Ca^2+^ signal transduction could lay a foundation for selective and drug-free treatment.

Here, we review how different optogenetic approaches have improved our understanding of Ca^2+^ signaling proteins and have helped to circumvent technical difficulties. Therefore, we first give a brief overview of the diversity of Ca^2+^ ion channels (Figure 1 and Section 2) and showcase in the following sections their specific use with regard to optogenetics (Section 3) and how these applications have advanced our knowledge of Ca^2+^-dependent physiological and pathophysiological processes (Section 4).

### 1.2. Ca^2+^ Entry Pathways

In general, Ca^2+^ entry mechanisms are classified into voltage-dependent and voltage-independent [19]. The first class is controlled by voltage-gated Ca^2+^ ion channels (VGCCs). These predominantly provide Ca^2+^ entry in electrically excitable cells [20,21] such as nerve and muscle cells, however there is emerging evidence that they are also essential in non-excitable cells, such as osteoblasts [22] and red blood cells [23].

The second class, voltage-independent Ca^2+^ influx, is mediated by ligand-gated ion channels, which can be divided into three groups: (i) Ca^2+^ depletion activated/store-operated Ca^2+^ ion channels (e.g., CRAC, TRP), (ii) store-independent Ca^2+^ ion channels (e.g., TRP, ARC), and (iii) Ca^2+^ influx via receptor channels (e.g., NMDA, P2X) [19]. Group (i) ion channels are activated by Ca^2+^ store-depletion. Group (ii) comprises a broad range of ion channels that can be activated by a wide variety of stimuli, among which some are also called receptor-activated or ligand-gated. Group (i) and (ii) are particularly crucial in non-excitable cells and fulfill a variety of physiological functions, including cell proliferation, migration, gene transcription and secretion [24,25,26,27,28]. Group (iii) ion channels, also termed ligand-gated ion channels, are especially important in the nervous system, but also excitable and non-excitable cells [29,30,31].

In the following, we provide a brief description of the above mentioned types of Ca^2+^ ion channels and their known activation mechanisms.

#### 1.2.1. Voltage-Gated Ca^2+^ Ion Channels (VGCCs)

The family of voltage-gated Ca^2+^ ion channels includes ten homologues, which are classified into three groups based on their different tissue expression and biophysical characteristics, each containing several isoforms, as follows: Cav1 (Cav1.1-Cav1.4; L-type), Cav2 (Cav2.1, N-type; Cav2.2, P/Q-type; Cav2.3 P/Q-type) (Figure 1A), and Cav3 (Cav3.1-3.3; T-type) [20]. They are activated upon membrane depolarization, which is sensed via positively charged amino acids in the fourth of six transmembrane (TM) domains per subunit. Furthermore, several auxiliary subunits regulate the function and trafficking of the L-, R-, P/Q-, and N-type pore complex, but not, or with less pronounced effects, those of T-type channels [20,32,33].

All subtypes are critical for the precise control of muscle contraction, neurotransmitter release, cell excitability and visual sensation. Dysfunction of ion channels can lead to diseases such as epilepsy, night blindness or neuropathic pain [34,35]. Furthermore, VGCCs seem to be related to the regulation of cancer cell growth as they are overexpressed in several cancer cell types [36,37] e.g., T-type channels in prostate cancer [38].

Despite their clinical relevance, current Cav channel inhibitors have severe disadvantages such as irreversibility, no spatial control, and side effects [39], therefore there is an explicit need for new approaches.

#### 1.2.2. Store-Dependent Ca^2+^ Ion Channels

Store-operated Ca^2+^ ion channels are activated by a biphasic Ca^2+^ signaling mechanism involving Ca^2+^ depletion from the endoplasmic reticulum (ER), which activates Ca^2+^ entry into the cell from outside [24]. Since the discovery of this link, the molecular components have remained unknown. It has been shown that Ca^2+^ entry is triggered by receptor stimulation at the plasma membrane, which initiates a signaling cascade leading to the production of phospholipase C (PLC), diacylglycerol (DAG), or inositol 1,4,5-trisphosphate (IP_3_). Some of the members of the transient receptor potential (TRP) ion channels are activated by IP_3_ or DAG, including, in particular, some TRPC channels. Thus, they were assumed to be involved in the formation of store-dependent Ca^2+^ ion channels.

Since only a few members of the TRP channel family are activated in a store-dependent manner, we give a separate brief description of TRP ion channels in Section 1.2.4 after the chapter on store-independent groups of ion channels.

In 2005/2006, the molecular components of the prototypic store-operated Ca^2+^ ion channel, the so-called Ca^2+^ release-activated Ca^2+^ (CRAC) ion channel (Figure 1B), were identified, namely the stromal interaction molecule 1 (STIM1) and Orai1, the Ca^2+^ ion channel [24,25,40,41,42]. The CRAC ion channel (Figure 1B) is composed of two transmembrane proteins, STIM1, the Ca^2+^ sensor in the ER membrane [40,41], and Orai1, the Ca^2+^ ion channel in the plasma membrane [42,43]. STIM1 contains a Ca^2+^ binding EF-hand at its ER-luminal N-terminus, which is connected to the C-terminus via its single TM domain. Its quiescent state is maintained by Ca^2+^ bound to this EF-hand and by a tight conformation of the helical regions in the C-terminus. Upon Ca^2+^ store-depletion, STIM1 loses bound Ca^2+^ and undergoes a conformational change that propagates from the N-terminus across the TM domain to the C-terminus. The C-terminal part extends and exposes sites that trigger STIM1 oligomerization and coupling to Orai1 [44,45,46,47,48].

Based on a variety of structures [49,50,51,52], the Ca^2+^ ion channel Orai1 forms a hexameric complex, which is unique among Ca^2+^ channels. In addition to Orai1, the Orai channel family contains Orai2 and Orai3 [42]. All Orai isoforms can form homo- as well as hetero-hexameric complexes [53,54,55]. While the homomeric Orai1 channel is sufficient to constitute together with STIM1 the CRAC channel [56], the heteromeric Orai assemblies can establish a set of store-operated Ca^2+^ ion channels, and thus can be responsible for a diversity of cellular Ca^2+^ signals, including sustained or oscillatory ones [57].

STIM1 couples to the Orai1 C-terminus at the periphery of the channel complex and triggers a conformational change within the Orai channel to establish pore opening. The latter necessitates a global conformational change of the entire Orai1 complex which involves a series of gating checkpoints to capture an opening permissive conformation, which has been reviewed by us in detail [58,59].

#### 1.2.3. Store-Independent Ca^2+^ Ion Channels

The discovery of the STIM and Orai proteins has opened up new research directions in the field of receptor-gated Ca^2+^ signaling. Given the prototypical activation mechanism and the importance of regulated Ca^2+^ entry, it is not surprising that Orai also forms a special channel complex. Stimulation of phospholipase C (PLC)-coupled receptors can activate not only the typical store-operated Ca^2+^ entry (SOCE) pathway via the CRAC channel but also the store-independent Ca^2+^ entry (SICE) pathway. The latter is activated by arachidonic acid (AA) or the AA metabolite leukotriene C4 (LTC4) and termed arachidonate-regulated or LTC4-regulated Ca^2+^ current (ARC/LRC). They are encoded by pentameric channel complexes consisting of distinct Orai isoforms, specifically three Orai1 and two Orai3 subunits [60]. These heteropentameric Orai1/Orai3 ion channels are particularly relevant in disease states, including cardiohypertrophy, injured smooth muscle cells, and breast, lung, and prostate cancer [61,62].

Furthermore, the secretory pathway Ca^2+^-ATPase, SPCA2, has been reported to establish store-independent, constitutive activation of Orai1 in breast cancer cells. Their association is mediated by both the N- and C-termini of SPCA2 [63]. Another store-independent activation pathway has been reported to occur via the small conductance Ca^2+^-activated K^+^ ion channels, in breast and colon cancer cells [64,65,66]. A potential link between Orai1 and SK3 has been suggested to be formed by the Sigma Receptor 1, a stress-activated chaperone [67]. Orai1 has been further reported to co-localize with the voltage-gated K^+^ ion channel Kv10.1, which promotes their activities [28,68]. However, the molecular underpinning establishing the latter two pathways and structural characterization of the diversity of store-independent Ca^2+^ entry pathways is still missing.

#### 1.2.4. Transient Receptor Potential Ion Channels

Notably, several TRP channel members are activated by a variety of other stimuli, and therefore form part of a store-independent or agonist-stimulated group of ion channels.

TRP channels represent a large and functionally versatile family of ion channel proteins that are permeable to cations. The first TRP protein gene was identified in Drosophila melanogaster [69] during the analysis of a fly mutant, whose photoreceptors showed only a transient response to persistent light stimuli. The TRP channel family includes 28 members, which are classified into six subgroups: TRPC (canonical), TRPV (vanilloid), TRPM (melastatin), TRPA (ankyrin), TRPML (mucolipin), and TRPP (polycystic). They share a common structure with four subunits, each composed of six TM domains, cytosolic N- and C-termini, and a pore-forming region between TM5 and TM6 (Figure 1C,D). Noteworthy, TRP channels may assemble as homo- and heterotetramers [70]. They are expressed in excitable and non-excitable cells, where they are involved in the regulation of cytosolic Ca^2+^ levels. Although most TRP channels conduct Ca^2+^, their selectivity for Ca^2+^ over Na^+^ can vary drastically. While TRPV5 and TRPV6 exhibit a P_Ca_^2+^/P_Na_^+^ ratio higher than 100, that of TRPM1 is less than 1. Among TRPV channels, TRPV5 and TRPV6 (Figure 1C) are the only highly selective Ca^2+^ ion channels. TRPC channels (Figure 1D) are generally non-selective cation channels. TRPA1 and among TRPM channel isoforms, TRPM3, TRPM6, and TRPM7 exhibit high Ca^2+^ conductance. Activation of TRP channels can lead to membrane depolarization or modulate the driving force for Ca^2+^ entry, thus making them essential in both excitable and non-excitable cells [71]. A unifying property of the very diverse TRP family is that the activation of each individual ion channel can be triggered by a variety of physical and chemical stimuli of both exogenous and endogenous origin [72]. In this review, among TRP channels, TRPV, TRPC, and TRPA1 are the main focus. While the activation of TRPV1-4 can be triggered by heat, by chemical, or, for some, by voltage [73,74,75], TRPV5 and TRPV6 are highly Ca^2+^ selective, constitutively active TRP channel types [70,71]. TRPC channels can all be activated in response to PIP_2_ hydrolysis [76,77]. Some have been reported to activate upon store-depletion, such as TRPC1, TRPC3, and TRPC4. Other TRPC members (e.g., TRPC6, TRPC7) are supposed to be activated by receptor stimulation, but in a store-independent manner [78,79,80]. TRPC channel modulation is even more complex, as some of the TRPC members, specifically TRPC1, TRPC4, and TRPC5, have been reported to directly interact with STIM1 [77,81,82]. TRPA1 channels are temperature sensitive, voltage sensitive, and respond to a variety of reactive and non-reactive drugs, including, for instance, cinnamaldehyde or nicotine, respectively [83].

TRP channels are found in virtually all cell types and tissues and are essential for a variety of physiological processes, including sensory (e.g., taste transduction, nociception, and temperature sensation), homoeostatic (e.g., Ca^2+^ and Mg^2+^ reabsorption), and motile functions (e.g., muscle contraction) [70]. Hence, it is not surprising that a range of pathophysiological processes, including pain, inflammation, and cancer, can be linked to the malfunction of a TRP channel member, either due to defects in the TRP gene, altered channel expression, or abnormal communication with the environment [38,70,84,85,86]. TRP channels are therefore important targets for the potential therapy of disease; however, only a handful of therapeutically useful drugs have been developed. To date, only TRPV1 has been targeted by a drug for the treatment of pain and bladder and gastrointestinal disease [70,87].

#### 1.2.5. Receptor Ca^2+^ Ion Channels

For Ca^2+^-permeable, ligand-gated Ca^2+^ ion channels, including, for instance, the purine ionotropic P2X, the cys-loop and the ionotropic glutamate receptor (iGluR) families, play an essential role in signal transmission in the nervous system as well as in synaptic plasticity [88].

iGluRs are ligand-gated cation channels that are categorized into three broad, structurally distinct functional classes: the α-amino-3-hydroxy-5-methyl-4-isoxasolepropionic acid (AMPA) receptors (Figure 1E), kainate receptors, and NMDA receptors. Despite the diversity, iGluRs form tetrameric complexes of identical or similar subunits. They contain a transmembrane domain segment, which is linked at the extracellular side to a ligand-binding region, and an amino-terminal domain that regulates their assembly, trafficking, and function. They are activated by the neurotransmitter glutamate [89]. In the conductive state, iGluRs are non-selective to cations, often also Ca^2+^, driving membrane excitation. Whether Ca^2+^ is allowed to pass depends in particular on the iGluR subunits, their arrangement in the channel complex, and various auxiliary subunits. It is becoming apparent that Ca^2+^ influx via GluRs plays a significant role in synaptic plasticity and thus memory and learning. Moreover, NMDA receptors have been identified to facilitate the release of glutamates via Ca^2+^ enhancements at the synapses and to participate in neuron-glia communications [88,90]. NMDA receptor dysfunction, both hypofunction and hyperactivity, can be responsible for the development of a variety of nervous system disorders, including ischemic brain injury, pain, depression, chronic neurodegenerative diseases, and schizophrenia [91,92]. There is emerging evidence that dysregulation of glutamate receptors can even lead to malignant cell growth [93].

P2X receptors are a family of seven subunits (P2X1–P2X7) that form homo- or heteromeric assemblies and function as cationic ATP-gated channels. Their multiple combinations have given rise to a range of P2X receptors with distinct Ca^2+^ permeability, kinetics, and pharmacological sensitivity. They are widely distributed in neurons, but also in non-neuronal cells, including, immune, glial, epithelial, and smooth muscles cells. There is evidence that Ca^2+^ entry via P2X receptors contributes to synaptic and neuromuscular transmission and neuromodulation; it is part of the cardiovascular system, the immune system, as well as secretion [88,94]. Alterations in the P2X receptor expression are responsible for a variety of pathological conditions in the brain and special senses, as well as of the cardiovascular, respiratory, gastrointestinal, and urinogenital systems [94]. Since ATP is a widespread and abundant factor in the tumor environment, it is not surprising that P2X receptors are important regulators of tumor growth [95].

The Cys-loop family is another group of transmitter-gated ion channels, including some more anion-selective (GABA and glycine receptors) and some more cation-selective (acetylcholine (ACh) (Figure 1F) and 5-HT3 receptors) receptors. All Cys-loop receptors have a common pentameric structure, with the pore in the middle of the five subunits. Each receptor complex consists of an extracellular segment, the transmembrane part, and an intracellular domain [96]. Ca^2+^ influx via ACh receptors contribute to neuronal differentiation, synaptic plasticity, and cholinergic modulations of transmitter release in the brain. It can also modulate neuronal signaling via Ca^2+^ dependent signaling cascades involving PKC, PKA, or CaMKII [88]. Pathogenic roles of Cys-loop receptors include neurodegenerative diseases, muscle dysfunctions, autoimmune response, and psychiatric disease [97]. Both muscarinic and nicotinic ACh receptors have been associated with numerous carcinomas [98].

Despite initial setbacks, new clinical agents based on glutamate and acetylcholine receptors are making promising progress. With these advances comes the very real prospect of breakthrough drugs that will expand treatment options for many patients. Until then, however, there are still several limitations to overcome, such as a better understanding of the structure–function relationship of these receptor channels and optimizing target specificity and undesirable side effects [99,100].

Summarized, the spectrum of Ca^2+^ ion channels governing a variety of physiological and pathophysiological processes is very broad. To intervene in ion channel function and associated downstream signaling events in terms of therapeutic strategies, a detailed understanding of the gating and regulatory mechanisms of Ca^2+^ ion channels is required. To date, traditional experimental methods such as molecular biology, biochemistry, electrophysiology, fluorescence microscopy, structural biology, and pharmacology have provided a plethora of insights into ion channel mechanisms. Unfortunately, these techniques lack sufficient temporal, spatial, and structural resolution, limiting our understanding and selective modulation of ion channels. Therefore, technologies with high spatiotemporal resolution that are sufficiently fast to synchronize with the dynamics of the ion channel in action during healthy and pathological states are urgently needed. Optogenetics has emerged as a powerful tool to meet those criteria. In the following, we provide an overview of the optogenetic tools used in the field of Ca^2+^ signal transduction to obtain precise control over the activity and mechanisms of ion pore complexes, as well as the new mechanistic insights gained as a result.

## 2. Optogenetics to Unravel the Mechanism of Ca^2+^ Ion Channels

Light is a suitable tool to obtain highly accurate control over biological processes. Various optogenetic technologies harness the power of light by transferring light-sensitivity to ion channels using light-sensitive proteins or chemical compounds which are described in detail in the following sections. Optogenetics involves the combination of optical and genetic methods to achieve high spatiotemporal control over single cellular events using light [101] and has evolved significantly over the last decades. It originally emerged from neuroscience to monitor the activity of neurons over a range of time scales [102]. Now, this revolutionary technology is used to control diverse cell types and is gaining popularity in biomedical research beyond neuroscience.

Regarding Ca^2+^ signaling pathways, optogenetics offers the unique opportunity to generate various Ca^2+^ signals, either sustained or oscillatory ones, by controlling the applied light intensity and pulse, which enables the qualitative and quantitative control of cellular Ca^2+^ levels. This is crucial to obtain precise control over individual events in the cell independent of cellular upstream processes [103,104,105].

### 2.1. Light Sensitive Chemical Tools

Chemical molecules that have so far been implicated in the light-controlled modulation of Ca^2+^ signal transduction processes in the cell are either caged groups or photoswitchable ligands (PLs) (Figure 2).

Caged compounds (Figure 2A) contain a photoremovable protecting group, which keeps a molecule (ATP) or biologically relevant element, such as the Ca^2+^ ion, from being active. Only when irradiated with UV light can the caged group be cleaved off, and the biologically active molecule or element is then released [106]; they are also advantageous in transferring light-sensitivity to endogenous proteins in native tissues; however, they lack target-specificity [106].

Photoswitchable compounds (Figure 2B,C) usually contain azobenzenes that switch between two conformations, trans and cis, upon irradiation with light of alternating wavelengths. Thus, photoisomerization can modulate or alter protein function depending on the wavelength. These azobenzenes are typically decorated with the agonist or antagonist of a receptor to selectively confer photosensitivity. Among the diversity of PLs, some are used as soluble molecules, while others are bound to the target molecule. The latter require genetic modification of the target protein to allow covalent binding of the ligand, for example via disulfide bridges. Currently, there are a variety of modes of action of photoswitches through which they can control protein action. These include, for instance, varying the distance between a drug and its binding site, inducing lateral pressure in the membrane or exerting a pincer-like motion on the channel gate [106,107].

Various chemical modulators that act either indirectly or directly on the function of ion channels are described in the following.

#### 2.1.1. Light-Sensitive Chemical Tools That Modulate Cytosolic Ca^2+^ Levels

Light-sensitive Ca^2+^ mobilizers were developed in early approaches to directly generate Ca^2+^ signals with controlled ON- and OFF-time and to study their effect on cellular function. Caged Ca^2+^ variants (Table 1) were extensively used for the direct modulation of the cytosolic Ca^2+^ concentration, which improved the kinetic resolution compared to conventional approaches. Fundamentally, the Ca^2+^ ion is tightly confined in metal chelators under dark conditions [108,109,110,111,112], while it is released upon UV light irradiation. This generates Ca^2+^ spikes in the cytosol, which can trigger Ca^2+^ signaling processes in the cell [108,113,114]. Conversely, Ca^2+^ reactions can be blocked by photoactivatable Ca^2+^ scavengers (e.g., diazo-2, diazo-4) that chelate free Ca^2+^ in the cytosol [115,116,117,118,119]. With these caged molecules, the disadvantages of conventional methods can be overcome and thus spatiotemporal control of intracellular Ca^2+^ signals and levels can be achieved [115,120,121,122]. However, prolonged UV light irradiation leads to the local depletion of these caged compounds due to by-products released after photolysis. Other disadvantages of stimulation by UV light in vivo, besides irreversibility, include a deleterious and toxic effect on tissue, limited depth of penetration into tissue and low delivery efficiency [108,123,124].

#### 2.1.2. Light-Sensitive Chemical Tools Interfering with Ca^2+^ Signaling Cascades

Caged molecules, such as IP_3_ [120,131,132,133,134,135] or adenosine triphosphate (ATP) [122,136] enabled indirect control of cellular Ca^2+^ signaling pathways [137,138,139] (Table 1). IP_3_ triggers the release of intracellularly stored Ca^2+^ and the subsequent store-operated Ca^2+^ influx into the cell [134,135]. Conversely, ATP lowers the Ca^2+^ concentration by activating multiple Ca^2+^ pumps (e.g., PMCA, SERCA, Ca^2+^-ATPase) [140] Caged IP_3_ was first synthesized in the late 1990s [132] as a cell membrane-diffusing chemical source of exogenous IP_3_. Similarly, caged ATP was first introduced in 1978 by Kaplan et al. [122]. Uncaging by UV light application leads to free IP_3_ or ATP and, subsequent, initiations of related signaling cascades by binding to membrane-localized receptors. Moreover, the neurotransmitter glutamate [137,138,139] as well as several receptor channel agonists and antagonist were caged [141,142,143,144,145,146,147]. Uncaging events were used to gain temporal control over the subsequent processes, thereby making it easier to study them in a controlled environment [122,148].

#### 2.1.3. Light Sensitive Chemical Tools That Interfere with Ca^2+^ Permeable Ion Channels


*Photochromic Ligands Acting on Voltage-Gated Ca^2+^ Ion Channels*


For voltage-gated Ca^2+^ ion channels, only a few PLs have been reported so far. It is worth noting that they usually do not act selectively on Ca_V_ channels, but also on several voltage-gated Na^+^ and K^+^ ion channels. In particular, a symmetrically substituted azobenzene with a quaternary ammonium ion on each side, termed QAQ (Figure 3A, Table 2), modulates Ca_V_s in addition to several other voltage-gated ion channels [149,150,151]. Interestingly, this QAQ compound acts only on the inside (Figure 3A), but not on the outside of the ion channel [149]. Since this compound is membrane-impermeable, it must be transported into the cell. Remarkably, this is possible via TRPV1 or P2X channels while their pore is dilated. This allows for the selective targeting of cells expressing these types of ion channels [149,150].

In addition, a permanently charged, light-switchable triethylammonium ion AzoTAB (Table 1) is currently available to block voltage-gated Ca^2+^ ion channels [152,174]. Despite the knowledge that these photochromic ligands act from the intra- rather than the extracellular side, an understanding of their mechanistic effects on the ion channel structure and their binding sites is still pending. Moreover, this would help to optimize available light-sensitive ligands in terms of their target-specificity for a particular type of voltage-gated Ca^2+^ ion channel.


*Photosensitive Ligands Acting on TRP and CRAC Ion Channels*


Among members of the TRP channel family, photosensitive ligands (Table 2) are currently available for TRPV1, TRPV6, TRPA1 and TRPC3 (Figure 3B).

For the photopharmacological modulation of TRPV1, photoswitchable agonists/antagonists (AC-4, ABCTC) were generated by incorporating an azobenzene into the structure of TRPV1 agonists. AC-4 was able to antagonize the action of the known TRPV1 agonist capsaicin. It is of mechanistic relevance that voltage-dependent activation is inhibited by trans-AC-4, whereas capsaicin-mediated activation is impaired by cis-AC-4. In contrast, ABCTC acted as a cis-antagonist on voltage-activated TRPV1, while capsaicin-activated TRPV1 currents were unaffected [154]. These findings suggest that these photoswitchable compounds might act at different sites in the channel complex depending on the mode of TRPV1 activation. Although it is hypothesized that AC-4 and ABCTC act in or near the capsaicin binding site identified by cryo-EM, further studies are needed [87,175,176]. In addition, the endowment of the agonist capsaicin with photoswitchable fatty acids enabled the development of azo-capsaicins (e.g., AzCa-4) that provided reversible optical control of TRPV1. Given the experimental evidence that capsaicin completely abolished the effect of AzCa-4 and the structural similarities of AzCa-4 and the vanilloid TRPV1 agonists, it is tempting to speculate that this azo-capsaicin binds to the capsaicin-binding pocket of TRPV1 (Figure 3B) [87,153].

Regarding TRPV6, the first photoswitchable ligand based on a TRPV6 inhibitor appended with a phenyldiazo, was reported to allow rapid photoswitching. Specifically, it blocks TRPV6 currents when it captured the cis-state. Mechanistic insights into the action of this light-sensitive compound are still pending (Figure 3B) [177].

With respect to TRPA1, a photoswitchable rhodanine derivative, optovin, was synthesized for reversible and cyclic control of its activity in the msec to sec range. The proposed mechanism is that optovin triggers the activity of TRPA1 via a light-activated intermediate state. This state is reversibly formed by the generation of the covalent thioether with redox-sensitive key cysteines (Cys 621, 633, and 856) in the channel (Figure 3B). Interestingly, light-induced TRPA1 activation was rapidly reversed in the dark in the presence of optovin, but the exact mechanism of deactivation remained unclear [155,156].

Moreover, some members of the TRPC channel family (e.g., TRPC2, TRPC3, TRPC6) are activated by diacylglycerols (DAGs). While investigations with DAG did not provide significant insights into the gating mechanisms of TRPC channels, photosensitive ligands helped to overcome these hurdles. Photoswitchable diacylglycerols (PhoDAGs, OptoDArG) containing photoswitchable fatty acids proved valuable for the precise optical manipulation of TRPC2, TRPC3, and TRPC6 activity [157,158,178]. The use of OptoDArG, a highly efficient photochromic ligand for TRPC3, provided novel mechanistic insights into TRPC3. Remarkably, OptoDArG facilitated the activation of a TRPC3 mutant containing the G652A substitution near the selectivity filter. Since classical pharmacological studies have already indicated that G652 is involved in lipid sensing, together with the findings on its light-mediated lipid modulation, it was concluded that G652 is a pivotal element in the recognition of lipid mediators in the channel’s vicinity (Figure 3B). Whether this occurs in a direct or indirect manner requires further investigation [87,158].

Recently, a library of PLs was synthesized for CRAC channels, the prototype among store-operated Ca^2+^ ion channels. For this purpose, known CRAC channel inhibitors GSKs [179,180] and Synta66 [181,182] were converted into several photoswitchable derivatives, namely piCRAC (Figure 3C). In particular, piCRAC-1, based on the GSK-5498A compound, was able to reversibly switch CRAC channels on and off by light stimulation at alternating wavelengths. Specifically, the cis-state of piCRAC-1 enabled the inhibition CRAC currents. Recently, Synta66 was reported to interact with a region near the pore [183]. Therefore, it is tempting to assume that CRAC channel GSKs and piCRAC-1 act near the pore (Figure 3C), but further evidence is still needed [159].


*Photochromic Ligands Acting on Ca^2+^ Permeable Receptor Ion Channels*


Among Ca^2+^ permeable receptor channels, a huge library of PLs (Table 2), either soluble or tethered ones, are already available.

For AMPA receptors, the light-switchable ligands containing an azobenzene together with a tetrazolyl (ATA) or quinoxaline-2,3-dione derivative (ShuBQX-3) have been shown to be efficient for their reversible control. Among ATA ligands, MD simulations demonstrated that ATA-3 reversibly binds to the ligand-binding domains within AMPA receptors, leading to the rupture and reformation of hydrogen bonds [160,161]. ShuBQX-3 is thought to bind to the ligand-binding domain of the glutamate receptor [162].

Among kainate receptors, the iGlu subunits can be reversibly activated and inhibited with light by means of a soluble photoswitchable ligand, namely GluAzo (Figure 3D) [163,164,165]. To gain structural insight into the action of the GluAzo photoswitch, a crystal structure of the ligand-binding domain of the kainate receptor was solved together with the GluAzo switch. This structure showed that the binding pocket of the glutamate moiety of GluAzo is analogous to that of soluble glutamate (Figure 3D) [184]. Furthermore, this structure contributed to the understanding of the structural changes in the glutamate binding of the GluAzo switch in both the trans or cis state, which is valuable for the design of novel glutamate receptor photoswitches, as well as for future drug design.

Moreover, a library of covalently linked glutamate switches containing maleimide-azobenzene groups (L-MAG) was developed, leading to the design of the light-switchable kainate receptors (LiGluR, LiGluN). These compounds include photoswitches with shorter and longer linkers, versions with red-shifted properties, and two-photon reactive ones [185,186,187,188,189,190,191,192,193]. They were localized on their surface in close proximity to the ligand-binding domain within the kainate receptor variants. The opening of the pore occurs in an allosteric manner. L-MAG induced activation in either the cis or trans conformation, regardless of which position near the glutamate binding site was chosen for binding. These MAG variants thus provided new mechanistic insights into the function of the native receptor and helped to elucidate the gating kinetics and desensitization mechanisms of this receptor. Different point mutations allowed L-MAG to act as either a cis or trans activator, enabling selective activation of different subsets of receptor-expressing cells [194].

PLs (ATG, PNRA) have also been developed and successfully used for NMDA receptors [195]. Similar to LiGluR, NMDA receptors were made with cysteine substitutions bound to PLs (L-MAGs). Depending on the location of the introduced cysteine, the photoswitchable tethered ligands (PTLs) resulted in light-mediated agonism or antagonism [170].

A variety of photosensitive ligands, including soluble and tethered ones, have been advanced for ACh receptors (AzoCharCh, azo-PTA, bisQ, QBr). Specifically, an azobenzene choline functions as a photoswitchable ligand [196], while two others can be tethered to cysteines, MAACh (maleimide-azobenzene-acylcholine), and MAHoCh (maleimide-azobenzene-homocholine). Molecular docking studies have shown that the latter two are able to reach the ligand-binding site of the ACh receptor to allow photoswitchable gating (Figure 3D) [173].

Apart from photoswitchable ligands, the incorporation of photo-responsive unnatural amino acids has been accomplished to transfer light-sensitivity to glutamate receptor variants [197,198,199]. Depending on the site of insertion, light-mediated activation or inactivation has been achieved, which left the action of glutamate unaffected [197]. This method led to the novel mechanistic insight that AMPAR activation is likely accompanied by selectivity filter gating [198].

#### 2.1.4. Advantages and Disadvantages of Photochromic Ligands

A variety of light-sensitive chemical tools have been generated and developed. Optical control through the use of caged compounds is a valuable approach to influence cell physiology. However, the application of these compounds is limited by the structural diversity, physical and photochemical properties, and the photolysis efficiencies of the available caging groups [200,201]. The most challenging criterion is the incomplete inertness of the caged molecule in the absence of light, which can lead to off-target effects [106]. Furthermore, the spatial resolution of these compounds may be compromised due to the rapid diffusion of the caged compounds. At this point, it is worth mentioning that the application of the photocaging techniques in whole bodies of animals is still challenging and restricted to transparent model organisms such as zebrafish embryos [202]. The irreversible photoreactions, the low cellular delivery capability, the high concentration requirement, and the limited penetration within the deep tissue are the prominent obstacles for in vivo applications [121,123,203]. Thus, it is still required to expand the repertoire of caged compounds to optimize their application in biological systems.

The PLs, unlike caged compounds, can be switched between two configurations with temporal precision and can act selectively on a protein of interest [163,204]. Remarkably, photoswitchable ligands have advanced the optical control and enabled direct manipulation of native proteins. However, their specificity and efficacy in targeting a specific molecule are not high, as they have a small size and can therefore diffuse easily in the tissue [205]. In this aspect, PTLs represent a novel generation of light-regulated drugs, where the integration of absolute specificity of genetics tools improves the precision of regulation at the molecular level [201]. In particular, PTLs have a high local concentration at the target sites, which significantly improves the kinetics of photocontrol. This unique feature is achieved by minimally altering the protein of interest [106]. One of the notable limitations of the PTL approach is that it is restricted to solvent-accessible sites to guarantee proper conjugation with the protein, typically extracellular sites, leading to the exclusion of the transmembrane domains and intracellular sites [106]. Moreover, genetic manipulation is associated with specific limitations, such as immune responses and the disruption of normal physiology [206]. In general, the major drawback of most currently available PLs and PTLs is that ultraviolet light is required for at least one of the photoswitching directions, which reduces their biocompatibility and increases their phototoxicity [189].

Remarkably, there is currently a number of PLs that have been successfully used to confer photosensitivity to a variety of ion channels. This is of great importance to achieve precise control over physiological and pathophysiological processes, as is explained in Section 4. Furthermore, photosensitive ligands have been applied to some ion channels to gain valuable insights into the structure–function relationship of the ion channel. However, mechanistic insights are limited because the action of PLs is typically restricted to the extracellular side. Therefore, further tools are highly awaited.

### 2.2. Light-Sensitive Genetically Encoded Tools

Previously introduced light-sensitive chemical modulators either act from the extracellular side or are able to permeate across the cell membrane to interfere with intracellular signaling steps. In contrast, genetically encoded Ca^2+^ modulators (Table 3) must be introduced into the cell to be expressed. There are two main types of photoresponsive, genetically encoded tools which occur naturally: the opsins and photosensory domains.

#### 2.2.1. Opsins in Ca^2+^ Signaling

Optogenetic actuators are proteins that can transfer light sensitivity to a cell in which they are expressed [101,207,208,209]. For this purpose, nature provides a variety of light-sensitive proteins. The first identified powerful actuators in the era of optogenetics are opsins (Figure 4A), which are native photosensitive proteins with seven transmembrane domains. They can be categorized into two groups: the microbial opsins (Type I) and vertebrate opsins (Type II). Type I opsins are found in archaea, bacteria, and prokaryotic/eukaryotic microorganisms. They include the bacteriorhodopsin, a proton pump, the halorhodopsin, a chloride pump, and the channelrhodopsins (ChRs), which form non-selective cation channels [207]. Type II opsins are found in animal cells and are akin to the superfamily of G-protein coupled receptors (GPCRs), which detect the day–night cycle and control basic vision [210,211]. The light-sensitivity of type I and II opsins is due to a covalently bound retinal that forms a retinal–opsin complex. Exposure to light leads to isomerization of the retinal [101,207,208,209] and the subsequent activation of the opsin itself, leading to a proton–ion flux or the activation of the downstream signaling cascade, depending on the type of opsin. They are distinguished by their ability to interfere indirectly with ion channel function, either by altering the membrane potential (microbial opsins) or by triggering cellular signaling cascades (animal opsins) [212].

Channelrhodopsin-2 (ChR2) [213] was the first microbial opsin used to control the activity of neurons by illumination with blue light, which leads to depolarization and, consequently, to neuronal signal transduction [102]. To achieve the light-triggered modulation of cellular signaling pathways instead of depolarization, Airan et al. [211] developed the so-called OptoXRs (Figure 4B), which are light-sensitive rhodopsin/GPCR chimeras that specifically activate the cAMP or PLC pathway. The introduction of conserved residues of adrenergic receptors into the loop and C-terminal regions of G_t_-coupled bovine rhodopsin allowed not only the modulation of the concentration of specific second messengers, e.g., cAMP or IP_3_, by light, but also the activation of downstream signaling cascades. The latter included an increase in cytosolic Ca^2+^ concentration, making OptoXRs useful tools in the field of cellular Ca^2+^ signaling [211].

**Table 3 cells-10-03340-t003:** Summary of light-sensitive genetically encoded tools which are compared by their photosensitive module, their switching wavelength, their activation as well as deactivation time, and by the cell type/organism in which they were expressed.

Classification	Light-Sensitive Tool	Photosensitive Module	Wave-Length	t_1/2_ ON	t_1/2_ OFF	Organism/ Cell Type	Ref.
Opsin in Ca^2+^ signaling	Opto-XRs ^a^	rhodopsin/GPCR chimeras	504 ± 6 nm/dark	Optoß2AR = 2,1s, Optoa1AR = 125 ms	Optoß2AR = 3 s, Optoa1AR = 533 ms	HEK293 mice	[211]
	PA-CXCR4 ^b^	chimeric, photoactivatable rhodopsin-chemokine receptor	505 nm	3–20 min		HEK293 mice	[214]
	PACR ^c^	LOV2, AsLOV2	470 nm/dark			C. elegans HeLa	[215]
Photosensory proteins	opto-FGFR1 ^d^	AsCryptochrome 2 (CRY2PHR)	488 nm/dark	~15 sec	~40 sec	HeLa HUVECs	[216]
	opto-RGK ^e^	LOV2-SsrA/SspB	470 nm/dark			HEK293 C2C12 HL-1 cardiomyocytes	[39]
	OptoCRAC ^f^	LOV2(404–46)	470 nm/dark	23.4 ± 4.2 sec	24.9 ± 4.8 sec	HeLa	[105]
	Opto-STIM1 ^g^	CRY2(1–498)	470 nm/dark	64.5 ± 4.8 sec	274 ± 23.7 sec	HeLa	[104]
	hBACCS1 ^h^	LOV2 (404–538)	470 nm/dark	4.5-fold in [Ca^2+^] after 30 s		HEK293T	[103]
	hBACCS2 ^h^	LOV2 (404–538)	470 nm/dark	~30 sec	~180 sec	COS-7 HEK293 HUVEC hESCs astrocyte zebrafish embryo mouse hippocampus	
	dmBACCS2 ^i^	LOV2 (404–538)	470 nm/dark	~30 sec	~150 sec		
	dmBACCS2 NS ^i^	LOV2(404–538) (N425S)	470 nm/dark	~30 sec	~30 sec		
	dmBACCS2 ^i^ VL	LOV2(404–538) (V416L)	470 nm/dark	~30 sec	~480 sec		
	LOCa3 ^j^	AsLOV2 (404–546)	470 nm/dark	48.69 ± 4.53 s;~75 sec	56.84 ± 3.79 sec	HEK293	[217]

^a^ β2-adrenergic receptor/α2-adrenergic receptor; ^b^ chemokine receptor;^c^ Calmodulin; ^d^ FGFR and downstream signaling; ^e^ CaV channel; ^f^ STIM1 (aa 336–486); ^g^ STIM1 (aa 238–685); ^h^ STIM1 (aa 347–448); ^i^ STIM1 (aa 413–514); ^j^ Orai1 (Δ1–64 H171D P245T).

A similar approach was conducted by van Wyk et al. [218] by engineering chimeras using melanopsin, a Type II opsin, combined with intracellular portions of the metabotropic glutamate receptor mGluR6, to recover blindness in mice. This was possible as both melanopsin and mGlu6 are part of the GPCR family and have a highly conserved tertiary structure. With that, a light-sensitive Opto-mGluR6 was engineered to trigger hyperpolarization in retinal cells [218].

Chemokine receptors play a central role in cell migration, which is a fundamental process in various disease states such as inflammation and cancer [219]. To gain more precise control over these receptors, the unique shared structure–function relationships between photoresponsive rhodopsins and chemokine receptors were exploited. Optoimmunoengineering enabled the construction of a chimeric, photoactivatable rhodopsin-chemokine receptor, or the so-called C-X-C chemokine receptor type 4 (PA-CXCR4) (Figure 4B). With this distinctive optogenetic system, Ca^2+^ influx could be selectively controlled in PA-CXCR4 expressing cells, leading to a more precise understanding of chemokine-induced downstream signaling cascades [219], as described in Section 3.2.

Overall, opsins were the first optogenetic tools to gain precise control over biological events, but only at the level of single cells by changing the membrane potential or specific signaling cascades. In the meantime, the optogenetics toolbox offers a variety of other tools that allow for the precise control of protein activity.

#### 2.2.2. Photosensory Proteins

Non-opsin-based photosensitive proteins provide another opportunity for precise temporal and spatial control in optogenetics. Currently, two types of photosensitive domains, originating from the plant kingdom, are used as optogenetic tools to transfer photosensitivity to a protein of interest. These can be classified according to their function, distinguishing between photosensitive proteins that oligomerize or undergo a structural change upon irradiation with light. Their common feature is the binding of a chromophore as a cofactor for photosensitivity [220].

The first group of light-sensitive proteins that can oligomerize includes cryptochromes, phytochromes, and UV resistance locus 8 (UVR8). Cryptochrome 2 (*At*CRY2), originating from Arabidopsis thaliana, contains flavin adenine dinucleotide as a chromophore. CRY2 (Figure 4D) is activated by blue light, which triggers electron transfer and subsequent flavin reduction. As a result, the N-terminal photolyase homology region (PHR) of CRY2 undergoes a conformational change and can rapidly and reversibly homo- or hetero-oligomerize [221,222,223,224]. Heteromerization can occur together with the cryptochrome-interacting basic helix-loop-helix 1 (CIB1) [225,226,227]. Whether CRY2 homo- or heteromerizes depends on various factors, including their location in the cell, their orientation to each other, and the availability of the binding interface [226]. In contrast to cryptochromes, the phytochromes are sensitive to red and far-red light. These photosensitive proteins require the chromophore phytocyanobilin (PCB) to become light-sensitive. Upon irradiation with red light, it can switch between inactive cis- and active trans-isomerizations, which, in the case of the phytochrome B, triggers the interaction with the phytochrome-interacting factor PIF. This process is reversible by irradiation with far-red light [228].

The second group of photoresponsive proteins undergoes a light-triggered conformational change, thereby releasing the active site of a protein of interest. They are the light-oxygen-voltage (LOV) domains (Figure 4C) and belong to a family of blue-light sensing photoreceptors, the phototropins [229]. The LOV domain consists of a β-sheet complex, the Period-ARNT-Singleminded (PAS) core, that is coupled to the chromophore flavin mononucleotide (FMN) and a C-terminal Jα-helix [230,231,232]. In particular, the LOV2 domain, derived from the phototropin of Avena sativa (AsLOV2), is most commonly used [232,233]. After illumination with blue light, its Jα helix is exposed, which can release a protein-binding interface or a signal sequence [232,233] that enables protein–protein interactions or initiates cellular signaling cascades [103,234,235,236,237]. Aside from that, LOV domains have also been used to trigger multimerization of proteins. For instance, the improved light-induced dimer (iLID; Figure 4E), consisting of a LOV2 domain from Avena sativa and the E.coli peptide SsrA, was used to trigger the light-mediated release of the oligomerization domains SsrA and SsrB [238].

While these photosensory domains have been described in detail elsewhere [220,239], in the following, we focus on their application in the Ca^2+^ signaling field, where they have recently been used to transfer light-sensitivity to "blind" Ca^2+^ signaling proteins (Table 3). In addition, we discuss their utility to date, as well as some open questions for gaining new mechanistic insights into protein function.


*Genetically-Encoded Tools that Modulate Cytosolic Ca^2+^ Levels*


The use of the photosensory module LOV2 enabled the generation of the so-called photoactivatable Ca^2+^ releaser (PACR) (Figure 4F). PACR consists of the light-sensitive AsLOV2 domain, the Ca^2+^-binding protein calmodulin (CaM), and a Ca^2+^ stabilization domain, the M13 peptide [215]. Under dark conditions, PACR is a tightly packed complex that binds Ca^2+^. Irradiation of this fusion protein with blue light (450 nm) leads to the disassembly of the densely packed CaM and the M13 peptide, releasing bound Ca^2+^. This process can be reversibly repeated, which is a major advantage over light-sensitive chemical tools. Additional modification with cell compartment-specific labels (e.g., nuclear localization sequence) allows for the spatial localization of PACR [215].


*Light-Triggered Control of Receptor Activity*


Activation of fibroblast growth factor receptor 1 (FGFR1) has been hypothesized to be associated with Ca^2+^ entry and subsequent cell migration [240]. To better understand the relationship between these processes, an optically controllable FGFR1 (opto-FGFR1) [216] was developed. For this purpose, FGFR1 was linked to the photosensory protein and expressed in the human umbilical vein endothelial cells (HUVECs). The application of the opto-FGFR1 together with a membrane-bound Ca^2+^ indicator allowed the resolution of the development of Ca^2+^ signals upon stimulation by blue light. The use of different Ca^2+^ ion channel blockers showed that Ca^2+^ entry mediated by FGFR-1 was due to CaV channel activity [216]. This assay provided new insights into FGFR1-mediated cell migration, as described in Section 3.


*Photosensory Domains to Transfer Light Sensitivity to Ca^2+^ Ion Channels*


In the area of Ca^2+^ ion channels, photosensory domains have so far only been used to transfer light-sensitivity to the voltage-gated Ca^2+^ ion channels, Ca_V_1.2, and the CRAC channel (Figure 5). This enabled researchers to obtain high spatiotemporal control over protein function and downstream signaling events and, moreover, achieved a deeper understanding of the mechanisms, regulation, and function of the respective channels.


*LOV2 to Modulate L-Type Function*


The purpose of applying LOV2 to Ca_V_1.2 channels (Figure 5A) was to overcome the limitations associated with conventional Ca_V_ channel blockers. A four-member subfamily of Ras-like GTPases known as RGK proteins (Rad, Rem, Rem2, Gem/Kir) are negative regulators of Ca_V_ channels [241,242,243]. Optogenetic control of the inhibitory effect of REM, which occurs only upon its binding to the plasma membrane (PM), was achieved by using the optical dimerizer pair iLID (LOV2-SsrA) and sspB. This led to the generation of the opto-RGK tool [39]. The distal C-terminal Rem domain, which normally interacts with the plasma membrane, was replaced by sspB, whereas iLID was constitutively bound to the PM [39]. Blue light irradiation resulted in the acute recruitment of optoRGK to the PM and blockade of Ca_V_1.2 channels. The reversible CaV channel modulator, the opto-RGK tool, was expressed in several cell types, including HEK293 cells, C2C12 cells, and HL-1 cardiomyocytes. In all these cell lines, a pulse of Ca_V_-mediated Ca^2+^ influx is typically induced by KCl [39]. Using opto-RGK, a significantly attenuated KCl-induced Ca^2+^ influx was induced under blue light illumination. Whereas, under dark conditions, Ca^2+^ influx was restored [39]. Ca_V_ channels play a central role in the regulation of the nervous system, cardiovascular system, and various electrically excitable cells [244]. These proof-of-concept experiments of the opto-RGK have shown a potential therapy for heart disease treatment and several applications in the field of Ca_V_-related physiological processes.


*Light-Sensitive STIM1 Variants Based on Light-Induced Oligomerization*


As mentioned above, STIM1 is part of the CRAC channel, functioning as a Ca^2+^ sensor in the ER to activate the Ca^2+^ ion channel, Orai, upon store-depletion. STIM1 activation is initiated by the dissociation of Ca^2+^ from its luminal domain (EF-SAM). This triggers oligomerization and a set of structural alterations along the entire STIM1 protein [46,48].

The activation signal for this represents the aggregation of the N-terminal regions. Moreover, it is known that the cytosolic STIM1 C-terminal region is sufficient to activate Orai1 channels constitutively [245,246]. Thus, the idea arose to attach the cytosolic region of STIM1 to a construct, keeping it in the quiescent state. In the first attempt, the exchange of the STIM1 luminal Ca^2+^ sensing region by an FKBP/FRB-based chemical inducible dimerization system enabled the activation of STIM1 via rapamycin [247,248,249,250]. Additionally, it was shown that chemical crosslinking of the first cytosolic residue 233 of STIM1 leads to an extended state of STIM1 [251]. These successful approaches paved the way for the use of photosensitive multimerization domains to transfer light-sensitivity to STIM1. Here, optogenetics aims to reconstruct the molecular structure of STIM1 by installing genetically-encoded photoswitchable domains. These STIM1-based GECAs tools can reversibly and remotely mimic the principal steps of SOCE without the ER store depletion in both excitable and non-excitable cells.

In particular, the photolyase homology region (PHR-aa 1-498) domain of the optical dimerizer CRY2 was capitalized. It was linked to various STIM1 fragments to trigger STIM1 activation via light-triggered oligomerization instead of Ca^2+^ store-depletion in a reversible manner. Mechanistically, this strategy enabled the respective CRY2-STIM1 chimera to capture a quiescent state in the dark and an activated one upon irradiation with light [252,253]. This made it possible to study various CRAC channel activation steps, such as dimerization, PM localization, activation of Orai1 as well as inactivation time [104,105,250,254,255]. This idea was first realized by Kyung et al. [104] by fusing various STIM1 fragments of different sizes to the *At*CRY2 domain, thus containing the luminal N-terminus and the TM domain of STIM1 exchanged by the CRY2 domain. This so-called OptoSTIM1 (Figure 5B), consisting of the PHR domain of *At*CRY2 and the C-terminal STIM1 fragment aa 238-685, enabled the light-mediated activation of the Ca^2+^ influx of endogenous as well as co-expressed Orai1. While activation occurred within a minute, deactivation in the dark took roughly 4 min to accomplish. OptoSTIM1 was expressed in several cell types such as HEK293, HeLa, NIH3T3, HUVECs, human embryonic stem cells (hESCs), and astrocyte to trigger Ca^2+^ signaling via light. Furthermore, OptoSTIM1 can move alongside microtubules, shows punctae formation, ER-PM junction localization, and maintains the Ca^2+^ selectivity of Orai1 upon irradiation with blue light [250].

Recently, two novel variants of OptoSTIM1 with enhanced light-sensing properties were published: Monster-OptoSTIM1 (monSTIM1) [256] and enhanced OptoSTIM1 (eOS1) [257]. monSTIM1 carries the CRY2 mutation E281A, which enhances the quiescent state of the chimera in the dark, and an additional elongation of the C-terminus of CRY2 (A9-ARDPPDLDN) [258], increasing light sensitivity [256]. eOS1 represents an enhanced variant of OptoStim1, containing the CRY2 mutation E490G, which led to an increased oligomerization [253].

Alternatively, for the use of CRY2, light-mediated STIM1 oligomerization was achieved by heteromerization of the SsrA and SsrB domains. For that purpose, the improved light-induced dimer, iLID, was used [238]. While in the dark, SsrA is hidden by the AsLOV2 domain, and it is exposed upon blue light illumination. Subsequently, SsrA can dimerize with a SsrB domain [250]. In another strategy, STIM1 fragments were attached to CIB1, and the addition of soluble CRY2 molecules triggered STIM1 fragment activation upon irradiation with blue light [250].

Notably, the comparison between the application of Opto-STIM1 and Opto-CRAC (more detailed in the following subsection) in various cell types targeting the CRAC channel revealed that, while Opto-CRAC exhibited faster kinetics in boosting the Ca^2+^ entry, Opto-STIM1 showed higher quality in promoting Ca^2+^ influx. Furthermore, Opto-STIM1 shows large-scale applicability, and Opto-CRAC needs the coexpression of exogenous Orai1 to enhance Ca^2+^ entry into the cell [104]. Under the same experimental conditions with HeLa cells, about six times more Ca^2+^ repeats were obtained with Opto-CRAC than with Opto-STIM1 due to the higher kinetic activity. Moreover, transient Ca^2+^ signals with Opto-CRAC mimicked the physiologically-related Ca^2+^ oscillation in mammalian cells.

These optical STIM1-dimerizer tools are not only precious to obtain high spatiotemporal control over Ca^2+^ entry, but also provided novel mechanistic insights into the CRAC channel machinery. In particular, the CRY2-STIM1 homomerization tool was highly valuable to characterize the individual steps of the STIM1 activation cascade. The contact interface of CC1 and SOAR/CAD that mediates STIM1 autoinhibition was identified by attaching STIM1-C-terminal fragments of different lengths to CRY2 [250]. Increasing N-terminal deletions of CC1 revealed that the region aa250-342 is critical for maintaining the inhibitory clamp [250], in line with previous findings [259,260]. While the CRY2-STIM1-C-terminus chimera remained inactive before light stimulation, deletion of residues up to L251–L258 in CC1 resulted in constitutive activation already in the dark [250]. The exact determination of the counterpart region in the SOAR/CAD domain has not yet been clarified.

CRY2-STIM1 fragments containing STIM1 luminal or C-terminal fragments of different lengths were also valuable in deciphering key regions for STIM1 oligomerization. Light-induced, co-clustering experiments revealed that the ER-luminal SAM domain and the cytosolic SOAR region are vital factors for STIM1 oligomerization [250].

In addition, the determination of novel gain- and loss-of-function mutations as well as the characterization of disease-related mutants was facilitated by the use of CRY2-STIM1 C-terminus chimeras together with a screening of mutations generated by random mutagenesis [250]. However, further studies are needed to determine the reason for the functional changes in the identified mutants.

Moreover, CRY2-STIM1 chimeras helped to monitor the communication between STIM1 and the microtubule as well as the PM. In this context, it was shown that the TRIP motif within the C-terminus of STIM1 particularly interacts with the microtubule plus-end tracking protein EB1, while the positively charged polybasic cluster at the very end of the STIM1 C-terminus plays an important role in directing the activated STIM1 towards the ER-PM junctions to initiate STIM1-mediated activation of Orai1 [250].


*Light-Sensitive STIM1 Variants Based on Light-Induced Uncaging*


In another optogenetic engineering approach, the C-terminus of STIM1 was fused to the genetically encoded photoswitch domain AsLOV2. In the dark, the Jα-helix interacts strongly with the PAS core, in which this strict interaction ensures that STIM1 remains in the inactive state while the LOV2 domain masks the active site of the fused STIM1 C-terminus. Upon irradiation with blue light, the tight conformation of the Jα- helix and the PAS segment is released due to a conformational rearrangement of the chromophore. This exposes STIM1 C-terminus for coupling to the Orai1 channel [105,261]. While the first of these constructs exhibited a remarkable dark activity that could not be neglected, a deletion of CC1 reduced the dark activity. This is likely due to SOAR competing for coupling between the LOV2 and CC1 domains, thus reducing the caging efficiency [261]. This construct consists of LOV2_404-546_ and STIM1_336-486_, and has been termed Opto-CRAC. Its response to light occurs rapidly and reversibly with an activation time of 8 sec and a deactivation time of 20 sec [261].

Alternative LOV2-STIM1 chimeras, which exhibit comparable and high efficiency, are the so-called blue light-activated Ca^2+^ channel switch (BACCS) variants. The latter include hBACCS1, a fusion of LOV2 with STIM1 C-terminal fragment (aa 347-448), BACCS2, a dimer of BACCS1, and a corresponding Drosophila melanogaster variant (dmBACCS2) [103]. All of them showed comparable activation kinetics in Ca^2+^ imaging studies [103].

The most efficient light-sensitive AsLOV2 STIM1 chimera, known as BACCS (nlue light-activated Ca^2+^ channel switch), is composed of the minimal effector domain of STIM1 (347–448) fused to several truncated versions of the AsLOV2 C-terminus [103].

Mechanistically, LOV2-SOAR was suitable to characterize the prominent intramolecular interaction of CC1 and SOAR, maintaining the quiescent state of STIM1. For that, Ma et al. [250] exploited the knowledge that STIM1 truncated at 342, which contains the N-terminus, TM, and CC1 domain of STIM1, which couples to the soluble SOAR in the resting state, whereas it detaches upon store-depletion. Thus, the elegant assay composed of STIM1 1-342 and LOV2-SOAR was developed, and enabled to determine the relative interaction strength of SOAR to CC1 when attached to the ER versus Orai1 in the PM. In the dark, LOV2-SOAR remained localized in the cytosol, both in cells expressing only Orai1 or cells co-expressing both Orai1 and STIM1 1-342. Upon irradiation with blue light, LOV2-SOAR translocated to the PM due to coupling to Orai1. Interestingly, in cells containing both Orai1 and STIM1 1-342, at a 1:1 ratio, active LOV2-SOAR showed predominant ER localization [250]. This suggests that, on the one hand, SOAR on its own preferentially couples to CC1, and on the other hand, that additional forces are required to establish SOAR-Orai1 coupling [250]. For a more detailed characterization of the preferences of SOAR for either coupling to CC1 or SOAR, distinct ratios are still required to be tested. Furthermore, this approach is suitable to characterize the CC1-SOAR as well as SOAR-Orai1 binding interfaces.


*Light-Sensitive Orai Variants*


Besides light-sensitive STIM1 proteins, there are also light-sensitive Orai variants available. Since STIM1 and Orai1 are direct interaction partners, it was worth designing a chimeric construct of a light-sensitive STIM1 together with Orai1. Indeed, Ishii et al. [103] successfully generated fusion proteins consisting of a BACCS variant (hBACCS1, hBACCS2, dmBACCS2) and Orai1, either as separate proteins or linked together (named Orai1:BACCS2). Among all of those, the Drosophila form dBACCS2-dOrai represents the one with the best activation kinetics while inactivation kinetics can be strongly altered via two distinct point mutations; the best hBACCS variant is found in the dimeric form (hBACCS). Reversible activation of the BACCS variants is achieved via illumination with blue light and is abolished upon removal of extracellular Ca^2+^.

While this fusion protein resembles the original activation mechanism of the CRAC channel, the use of the LOV2 domain directly on Orai1 opened new possibilities. By incorporating the LOV2 domain in the flexible loop region between TM2 and TM3, known as loop2, and further rounds of optimization, including N-truncation and single point mutations, an STIM1-independent, light-activated Orai1 ion channel, known as light-operated Ca^2+^ channel (LOCa), was created. Specifically, the LOCa (Orai1 Δ1-64 _163_AsLOV2_164_ H171D P245T) exhibits robust light-mediated activation with low background activity in the dark [217]. This protein can be reversibly activated by blue light illumination. While the light-activated LOCa currents exhibit high Ca^2+^ selectivity and are blocked by a CRAC channel blocker BTP2, the currents are much smaller than those of overexpressed STIM1/Orai1 or GoF-Orai1 mutant currents. Furthermore, the characterization of several other CRAC channel features such as fast Ca^2+^ dependent inactivation, as reviewed in Krizova et al. [262], are still pending.

These light-sensitive Orai variants have been so far applied to modulate and study cellular downstream events and disease-related processes, as outlined in Section 3.2. However, they have not yet been used to gain mechanistic insight into Orai activation. Particularly, LOCa represents a promising tool to intensify our current understanding of the role of the loop2 region in Orai1 gating [263,264,265]. Interestingly, LOV2 integration only allowed to transfer light-mediated Orai1 activation in the presence of a constitutively active point mutation (P245T). The additional single amino acid substitution (H171Y) provides indications for the necessity of a correct interplay of cytosolic segments, as we recently demonstrated for the interplay of salt-bridge interactions within the cytosolic triangles [59,61].

#### 2.2.3. Advantages and Disadvantages of Naturally Occurring Photosensory Proteins

Opsins in Ca^2+^ signaling have proven to be valuable tools in controlling various biological events. However, one significant obstacle is the potential interference between the Ca^2+^-dependent and Ca^2+^-independent effects within the host cell.

By fusing the photosensitive domains to the protein of interest, the resolution was increased to an unprecedented subcellular level, and the precise control of the protein function was enabled by the intensity and duration of illumination. These elegant studies were able to mimic the dynamic properties of different biological cascades and have provided novel insights into the signaling network within the cell. It is also worth mentioning here that these tools have facilitated in vivo applications, as described in Section 3.2. For instance, the relatively small size (1–3 kb) of the LOV2 domain was successfully used with most of the available viral packaging systems [255], as they have a limited capacity to carry genetic material. In addition, the genetically encoded tools are more compatible for in vivo application as no external chromophores are needed to absorb photons [266]. One of the disadvantages is the incompatibility of certain light-sensitive domains with the FRET biosensors, for instance, the LOV domain and CRY2, due to the overlap of the blue light for their photoactivation with the GFP excitation wavelength [267]. Additionally, transferring photosensitivity to a protein of interest using photosensitive proteins requires extensive screening using molecular biology techniques to finally obtain a functional photosensitive protein. One of the noticeable limitations of this method is the location of the incorporation of the light-sensitive domain into the protein of interest; in particular, incorporation into transmembrane domains is not suitable. In transmembrane proteins, it is usually useful to incorporate these photosensitive proteins into cytosolic or extracellular segments near the functionally relevant sites.

## 3. Physiological and Pathophysiological Relevance of Novel Optical Tools in the Field of Ca^2+^ Permeable Ion Channels, Both In Vitro and In Vivo

The highly flexible optogenetics technology has developed rapidly over the past decade [102,107,268]. While optogenetic applications were originally performed primarily in neuronal systems, they have meanwhile been successfully employed in a wide range of other cellular systems, such as muscle and immune cells. Concerning Ca^2+^ dependent signaling, the high spatiotemporal control achieved by the diversity of optogenetic tools currently available, in addition to gaining mechanistic insights on Ca^2+^ ion channels, offers the possibility to precisely control or influence Ca^2+^ dependent physio- and pathophysiological processes within the cell, tissues, or even living organisms [106,107,124,209,254,255]. Below, we describe the milestones achieved in the understanding and manipulation of the cell physiology in vitro and in vivo using the variety of photosensitive tools in conjunction with Ca^2+^ signals and Ca^2+^ ion channels described above.

### 3.1. Light Sensitive Chemical Tools in Downstream Signaling and Physiology

#### 3.1.1. Caged Ligands

Among photopharmacological strategies, a variety of caged ligands have been successfully applied in the cellular physiology of neurons [269,270,271,272] as well as in muscle cells [109] linked to Ca^2+^ signaling. Accordingly, various caged compounds have been precisely generated and included in several areas of interest, which are highlighted in this section.


*Caged Ca^2+^*


In neurobiology, a photolabile Ca^2+^ cage (e.g., nitr-5, azid-1) was used to trigger neurotransmitter release, such as glutamate [113,125,273], which is critical in synaptic plasticity, to induce receptor stimulation and subsequent cellular signaling cascades. This application enabled to prove, for instance, that neurotransmitter release is not directly associated with a change in the membrane potential, but rather with elevations of intracellular Ca^2+^ levels [125]. Later, caged Ca^2+^ was successfully used to trigger glutamate release in astrocytes, which are considered to be non-excitable cells [272]. Furthermore, light-irradiation of photocaged Ca^2+^ was shown to activate K^+^ currents in neurons due to alterations of the membrane potential [274].


*Caged Glutamate*


In neurons in particular, caged glutamate combined with two-photon uncaging revolutionized the understanding of a variety of physiological events. For example, it was utilized to precisely determine the number of release events required to initiate forward-propagating action potentials in dendrites, known as dendritic spikes [129]. Furthermore, this optogenetic tool was employed to determine the Ca^2+^ dynamics in neurons, particularly in the dendritic spine, the small protrusion from the neuron’s dendrite. The sequence of synaptic activation, as well as the spine neck geometry, define the magnitude of Ca^2+^ signals, which are fundamental for brain function [275,276]. In addition to triggering Ca^2+^-dependent processes in neurons, the release of neurotransmitters through glutamate uncaging permitted to control the location of certain types of receptor ion channels, for example, in pre- compared to post-synaptic regions or subcellular compartments [277]. For AMPA receptors, a correlation of their individual densities with spine architecture and size in pyramidal neurons has been identified [138].


*Photolabile EGTA, photocaged IP3 and photocaged lipids*


In the muscular system, photolabile EGTA was successfully applied to induce light-mediated skeletal muscle fiber contraction [110]. In addition, the photocaged IP**_3_** enabled to show that only the contraction of smooth and skeletal muscle cells, but not of striated skeletal muscle fiber, is driven by IP_3_-dependent signaling pathways [120,130]. Photolysis of DAG was used to monitor the contractility of cardiomyocytes [278].

To study Ca^2+^ signaling in immune cells, caged IP_3_ was further used to show that the production of IP_3_ is sufficient to activate Ca^2+^ influx into T-cells [279]. Elevations in Ca^2+^ levels trigger a nuclear factor of activated T-cells (NFAT) transcription and subsequent secretion [280]. Secretion of cytokines and lytic-factors is linked to the reorganization of the microtubules.

Among published caged lipids, 1,2-DOG has been used to show that the dynein-dependent polarity of microtubules in primary T-helper cells is driven by DAG [281].

#### 3.1.2. Photosensory Ligands

The second set of optogenetics-based light-sensitive chemical tools includes many PLs, both soluble and tethered. These tools have been used to control TRP ion channels and ionotropic glutamate receptors [204] and have been applied in neuronal systems to reversibly trigger action potentials via light [186,204]. In this respect, their application has revealed valuable facts and provided a more profound understanding of several Ca^2+^ associated biological pathways in both in vitro and in vivo studies.


*In Vitro Applications of the Photosensitive Ligands to Unravel Ca^2+^-Dependent Processes*


Impressively, the TRP channels are attractive targets for the integration of several photoswitchable ligands and showed high flexibility in the application. Accordingly, (I) optovin and AzCA4 enabled the reversible activation of native murine TRPA1 [155] and TRPV1 [153] in dorsal root ganglion sensory neurons, respectively. (II) AzCA4 has been further applied in C-fiber nociceptors to activate TRPV1 [153]. (III) Photoswitchable diacylglycerols enabled optical control of TRPC2 and TRPC6 in tissue slices [158]. (IV) The first TRPC3/6 photoswitch enabled lipid-independent TRPC3 activation in hippocampal neurons and endothelial cells [282].

In addition to the previous, among ionotropic glutamate receptors in particular, photoswitchable ligand-gated kainate receptors (LiGluRs) have been used to establish the precise optical control of Ca^2+^-dependent processes, including exocytosis [283], neurotransmitter release [168], and glutamate release [166]. Moreover, different photosensitive and glutamate receptor forms (LimGluRs) enabled high spatiotemporal control over neurotransmitter release [169]. Moreover, the stimulation of the electrically silent cells, such as neurotransmitter release in astrocytes, was made possible by the utilization of particular LiGluR variants.

By extension, photoswitchable ligands tethered to N-Methyl-D-aspartate receptors (NMDRs) allowed the control of spine function and synaptic physiology over Ca^2+^ levels [170,284]. Specifically, an irreversible AMPA receptor antagonist (ANQX) was used to study AMPA receptor localization and trafficking and revealed that they are mainly transported to the extrasynaptic sites through lateral diffusion [285]. Two-photon uncaging was used to identify that AMPA receptors recruit to synapses of dendritic spines during long-term potentiation, a neuronal mechanism essential for learning and memory [286].

In addition to TRP ion channels and ionotropic glutamate receptors, the nicotinic acetylcholine receptor (nAChR) was also of interest, and thus a new caged nAChR agonist, ABT594, was developed. The local release of a nAChR agonist facilitated the identification of the types of neurons in which AChRs are concentrated. Light-induced activation was shown to be associated with enhanced cytosolic Ca^2+^ levels, supporting the hypothesis that AChRs conduct Ca^2+^ [143,144,287].


*In Vivo Applications of the Photosensitive Ligands to Unravel Ca^2+^-Dependent Processes*


The photochromic TRPA1 ligand (optovin) is, in addition to in vitro studies, a powerful tool that has been introduced into several biological systems. Particularly, it has enabled light-mediated control of dorsal fin movements [155], heart rate, and the pacing of human stem cell-derived cardiomyocytes of zebrafish [156]. Moreover, optovin proved suitable to activated TRPA1 in the ear of the mouse [155].

Moreover, small organic diffusible light-sensitive molecules (e.g., opto-PAMs and opto-NAMs of mGluRs), as well as photoswitches tethered to glutamate receptor variants (LiGluRs, LimGluRs), have been used for the light-mediated manipulation of zebrafish larvae [186,288,289,290]. Additional studies with LiGluRs have also been conducted with zebrafish, where the direct optical manipulation of intact nervous tissue provided an opportunity to discover specific areas and cell types accountable for novel functions and specific behavior [169,170,291,292].

Although more challenging, light-dependent neuromodulation has also been achieved in rodents with diffusible photosensitive mGluRs drugs [290,293,294]. Recently, cysteine anchored PTLs were used in vivo in the mouse brain (LiGluRs) [167], first in superficial layers and later in the deeper brain region (LinAchRs) [295].

Importantly, as the eye is a convenient target for drug delivery and has direct access to light, the introduction of PTL tools into the retinal ganglion cells (RGCs) was shown to be a promising approach to treat the retinal pathologies in blind mice to restore vision [296,297,298]. Of the various photoswitchable blockers, most are membrane-impermeable, except for photoswitchable quaternary ammonium blockers. These enabled pain control in rats [299] and restored visual function in blind mice [149].

### 3.2. Light Sensitive Genetically Encoded Tools in Downstream Signaling and Physiology

As mentioned earlier, optogenetics-based Ca^2+^ signaling has evolved rapidly. Remarkably, in addition to previous chemical tools, a variety of other instruments based on light-sensitive genetically encoded systems have been developed. In this section, we shed light on two main categories that can either be expressed in cells (Opsins) or fused with the protein of interest (photosensory domains). Both have been successfully exploited to precisely control various Ca^2+^-based biological activities and processes.

#### 3.2.1. Opsins

Historically, opsins first gained great importance in the neurosciences [209], but have since emerged as popular tools in non-neuronal tissues as well [300]. The most prominent microbial opsin, channelrhodopsin 2 (ChR2), induces strong membrane depolarization upon light illumination, which can trigger action potentials in neurons. With respect to Ca^2+^ signaling, traces of Ca^2+^ passing through channelrhodopsins enabled to activate non-neuronal cells, including glial cells [301], insulin-secreting pancreatic beta cells [302], immunologic, and endocrine-exocrine tissues [303]. Moreover, microbial opsins enabled the optogenetic control of mammalian brain tissue and freely moving mammals [209,303,304].

In addition, G-protein coupled photoreceptors (opto-XR, melanopsin) enable a broader targeting of cellular Ca^2+^ signaling pathways, both in neuronal as well as non-neuronal cells [211]. Naturally occurring melanopsins (originally found in retinal ganglion cells (ipRGCs)) were used to activate transient and stable Ca^2+^ signal generation to study Ca^2+^-dependent reactions in mammalian cells [305]. Indeed, melanopsin was employed to induce NFAT translocation essential for transgene expression [305]. This melanopsin-based linkage of these signaling steps, together with the implantation of light-inducible transgenic cells into diabetic mice, led to a robust increase in the glucagon-like peptide-1 variant (shGLP-1) after light-stimulation. This was able to mitigate glycemic excursion in mice with type II diabetes. In particular, several reports have suggested that shGLP-1 plays a critical role in glucose homeostasis modulation [306,307]. Therefore, this strategy could be suitable in the treatment of type II diabetes and the prevention of glucose-related pathologies.

In another application, light-triggered melanopsin was used to demonstrate that the decoding of NFAT depends not only on the oscillation frequency of Ca^2+^ signals, but also on the accumulated amount of Ca^2+^ during the oscillation period [308]. Furthermore, light-driven activation of melanopsin and downstream signaling could be used to generate local pacemaker activity in cardiomyocytes in embryoid bodies (EBs) derived from mouse ESCs [309].

Opto-mGluR6, the chimera of melanopsin and intracellular parts of mGluR6, was able to restore vision in mice suffering from photoreceptor degeneration. Moreover, this light-sensitive chimera not only restored the light sensitivity of retinal ganglion cells to ambient light, but also restored vision outside of the retina in the brain cortex [218]. On top of that, an attractive study helped to enrich our understanding of the molecular underpinning for cell migration using the chimeric rhodopsin-chemokine receptor, PA-CXCR4. This receptor allowed for optical the control of T-cell trafficking and migration under both in vitro and in vivo conditions. PA-CXCR4-expressing cytotoxic T lymphocytes exhibited light-activated directed migration or phototaxis. Integration of PA-CXCR4-expressing cytotoxic T lymphocytes (CTLs) into a mouse model of melanoma promoted intratumoral infiltration of CTLs and led to a rapid reduction in tumor growth [214]. This clearly underlines the therapeutic potential of this approach.

#### 3.2.2. Photosensory Domains

Extensive efforts have been invested in linking photosensory domains to Ca^2+^ signaling proteins to influence various Ca^2+^-dependent signaling pathways, which are described in the following.


*Light-Sensitive Receptor Opto-FGFR1*


The light-sensitive receptor, Opto-FGFR1 is not only a tool to trigger Ca^2+^ entry, but also to study the associated cell migration [240]. It was used to understand the link between Ca^2+^ wave generation and subsequent migration, which remained incomplete due to the complexity of these processes. In addition to opto-FGFR1 [216], light-sensitivity was transferred to the Rho small GTPase Rac1 (photoactivatable (PA-)-Rac1), which represents a downstream signaling node within the receptor tyrosine kinase (RTK) signaling cascade. These two optical tools provided a unique opportunity to monitor the individual steps of the signaling cascade from growth factor binding to migration independent of upstream processes. A comparison of cellular movement as a function of the two genetically engineered photosensory proteins revealed a more critical role of opto-FGFR1 over that of PA-Rac1 in cell reverse contraction. The Ca^2+^ influx associated with opto-FGFR1 via CaV channels generated a Ca^2+^ gradient in the migrating cells that increased from the front to the back. Using the light-induced activation of the PI3K cascade [310], it was shown that induction of Ca^2+^ sparklets requires the modulation of the membrane lipid by the PI3K pathway. Hence, the optogenetics toolbox has revealed new facts and an intrinsic function of local Ca^2+^ signals in directed cell motility [311].


*Light-Sensitive CRAC Channel Tools*


Since the CRAC channel is one of the major Ca^2+^ entry pathways in cells, it was an important target for several applications in physiological- and pathophysiological processes. Notably, by using optical activation of CRAC channels mediated by various photosensory domains, several light-sensitive tools have been generated (e.g., Opto-STIM1, Opto-CRAC, BACCS, eOS1, LOCa3), each with its own particular characteristics. In general, more direct control over downstream signaling, e.g., gene transcription, has been achieved. It is worth mentioning that, in addition to in vitro and in vivo applications, this chapter also highlights the extensive use of the light-sensitive CRAC channels, particularly in the deeper tissue layers of living organisms.

In cell physiology, proteins have been used to induce light-induced NFAT activation [103,104,105,217,257]. Enhancing the frequency of the applied light-pulse correlated with an increase in the extent of NFAT translocation to the nucleus [105]. Specifically, optogenetic CRAC channel tools led to a substantial luciferase/insulin gene expression [105]. In CD4+ T cells, Opto-CRAC triggered the production of signature cytokines, such as IL2 and IFN-γ. Similarly, human THP-1 macrophages expressing Opto-CRAC released IL-1β and processed caspase-1 after light stimulation, indicating the supporting role of the Opto-CRAC channel in promoting macrophage-mediated inflammatory responses [105].

More efficient gene expression was established by another powerful strategy based on the combination of the clustered regularly interspaced short palindromic repeats-associated-9 nuclease (CRISPR-Cas9) tools [312,313,314] with the genetically encoded photoactivatable Ca^2+^ actuators. This method allowed to precisely and reversibly control the CRISPR-Cas9 system and to avoid off-target effects. Here, a light-sensitive CRAC channel (Opto-CRAC) is the first leading player [217,315]. The second major player is the so-called calcium-responsive dCas9 fusion construct (CaRROT), consisting of the N-terminal fragment of the NFAT (residues 1-460) fused to dCas9 and transcriptional coactivators (VP64/VP160) [315]. This newly designed construct ensures that gene expression at the targeted loci does not occur without blue light illumination. This strict optical control is due to the phosphorylation of the NFAT fragment. Only light irradiation can initiate Ca^2+^ influx followed by cleavage of phosphate groups by calcineurin. The specificity of the system is enhanced by the absence of the C-terminal DNA-binding domain of the NFAT fragment, which could otherwise bind to endogenous NFAT targets [315].

Opto-CRAC channels were also incorporated into therapeutic dendritic cells in a mouse model of melanoma. Consequently, light-triggered Ca^2+^ responses in immune cells promoted the maturation and antigen presentation of dendritic cells. The latter enhanced T cell priming and activation, which facilitated melanoma destruction in these mice [105]. Bohineust et al. [257] additionally applied two-photon photoactivation, which offers the advantage of deeper tissue penetration both in vitro and in vivo. Specifically, eOS1 was transferred to the popliteal lymph nodes of mice to induce light-dependent elevations of Ca^2+^ levels. Furthermore, BACCS was suitable to induce light-mediated response in olfactory sensory neurons, as determined by electro-olfactograms.

Noteworthy, to get closer to the in vivo application of available optogenetic tools, lanthanide-doped upconversion nanoparticles (UCNPs) were used together with the Opto-CRAC channel. UCNPs work as nano-transducers and can emit light in the visible and ultraviolet range through near-infrared excitation. This unique feature enables the use of existing optogenetic tools in deeper tissue layers. The overlap of the emission spectrum of the UCNPs with the absorption spectrum of the LOV2 protein enabled their application in deeper tissue layers [105]. This made the successful wireless photoactivation of Ca^2+^-dependent processes in a mouse model of melanoma possible. Accordingly, antigen-specific immune responses were promoted and tumor growth and metastasis suppressed when illuminated with external NIR light.

Furthermore, the photoresponsive Orai1, LOCa3, was applied to various biological systems [217] to rescue Tet2 gene defect in hematopoietic stem cells and to partially rescue the Drosophila melanogaster of progressing neurodegenerative disease. CD8+ T cells expressing eOS1 allowed light-mediated control of motility and adhesion, as well as chemokine release [257].

Notably, Ishii et al. [103] found that their dmBACCS2::dORAI variant is unlikely to interact with endogenous mammalian variants and can therefore be used as an independent system with high spatial and temporal control to induce Ca^2+^ influx into the cell. Subsequently, the dmBACCS2-dOrai variant was used in osteogenic differentiation [316], as Ca^2+^ is essential for bone formation [317,318,319,320]. More specifically, Ca^2+^ spikes, or an overall increased intracellular Ca^2+^ concentration, could be stimulated by repeated blue light application, leading to an increased osteoblast differentiation [316].

The engineered Opto-STIM1 has been introduced into zebrafish embryos as well as in human embryonic stem cells (H9 cells), and has allowed successful light-triggered Ca^2+^ influx, which demonstrates its in vivo applicability. In this regard, Opto-STIM1 arose as a promising strategy to boost the learning capacity [104]. Principally, light-triggered Ca^2+^ influx led to an enhancement in the process of memory formation and subsequent contextual fear memory in mice. Moreover, monSTIM1 was shown to enhance fear-related learning behavior [256]. These findings point to the possibility of using optogenetics for immunotherapy and the regulation and enhancement of mammalian learning.

### 3.3. Conclusion on In Vitro and In Vivo Application of Optogenetic 

Collectively, this series of studies showed that the use of optogenetics in Ca^2+^-dependent signaling might be a promising procedure for future therapeutic applications. In addition, both light-sensitive chemical and genetically encoded tools are a real breakthrough for the study of various intracellular signaling pathways and are presently utilized in diverse research domains. They are characterized by overcoming the limitations of several conventional pharmacological methods, such as the breakdown of the kinetic conditions for the biological processes and the high speed of photoinduced biological signaling. However, several optimizations of their performances are still required, and various criteria should be considered before integrating them with the biological processes. These criteria include the absence of non-specific interactions within the physiological system, especially in the inactive state, and the improvement of the illumination conditions to minimize interference within cellular pathways or damage to the biological sample.

## 4. Perspectives

To date, numerous treatments have been developed for various diseases, including chemotherapy, hormone therapy, and biological therapy. However, many of these therapies employ drugs that are often harmful to the body, have limited specificity, and lack spatial resolution. A major drawback of these therapies is also the side effects, which can be worse than the disease itself. Therefore, there are various efforts to introduce optogenetics as the best alternative therapeutic tool to the existing methods to overcome the aforementioned obstacles. Furthermore, combining this unique strategy with the currently available therapeutic approaches may yield promising techniques to treat diseases more effectively and safely.

Although this strategy is still in the development stage, optogenetics has enabled its application in wide areas, such as cancer research, cardiovascular research, hepatology, retinal gene therapy, ophthalmology and others. A notable advantage is the excellent spatiotemporal control of the protein function that can be achieved with very short (millisecond) pulses of light. In this regard, the introduction of this unmatched temporal resolution into the biological system to control and investigate diverse Ca^2+^-dependent pathways would enrich the understanding of the mechanisms of different diseases and disorders due to the primary role of the Ca^2+^ ions in regulating cellular functions. However, while the optogenetic approach holds great promise for the future treatment of various diseases, there are still many questions that need to be addressed.

For example, how to safely deliver light-sensitive proteins into the human body and how to ensure high and efficient spatial expression into the particular cells or organs. Potential strategies represent the insertion of optogenetic tools, for instance, via viral vectors or the generation of the transgenic animal lines to express optogenetic constructs in a cell-type-specific manner. The newer methods are based on physical cues, such as electric field (electroporation), ultrasound waves (sonoporation), or laser pulse (photoporation) [321]. So far, all the treatments are limited to preclinical models. However, a holistic approach to get closer to human applications would be to combine optogenetics with the CRISPR-Cas9 system. This could allow the application of this system for different cell models. If this strategy works successfully in humans, it would be, on the one hand, a powerful technique to investigate physiological functions, and, on the other hand, the first drug-free therapeutic tool. Furthermore, an important consideration for medical use of optogenetics is the decision for the right wavelength and methodology of stimulation. Only wavelengths in the near-infrared range can penetrate deep into the tissue, however optogenetic applications are still rare. Among methods for light sources, for instance, LED, micro-LED, or laser are in development [322,323].

In addition, optogenetics has been shown to provide new mechanistic insights into protein function, which could be valuable for characterizing new targets for potential future therapeutic approaches. However, as outlined in this review, some unanswered questions regarding mechanistic findings have also remained, suggesting that further efforts and methods are needed to answer outstanding questions.

It is worth noting that the optogenetics technique paved the way for the development of the optoproteomics approach [324]. This novel approach is taking advantage of the genetic code expansion technology to widen the repertory of the 20 naturally occurring amino acids, which proteins are composed of, by many novel chemically synthesized unnatural amino acids (UAAs) [324,325]. This promising approach gives optoproteomics, as opposed to optogenetics, a broader meaning and consequently provides a unique opportunity to employ optical tools to study mediated biological functions in addition to site-specific proteomics.

## Figures and Tables

**Figure 1 cells-10-03340-f001:**
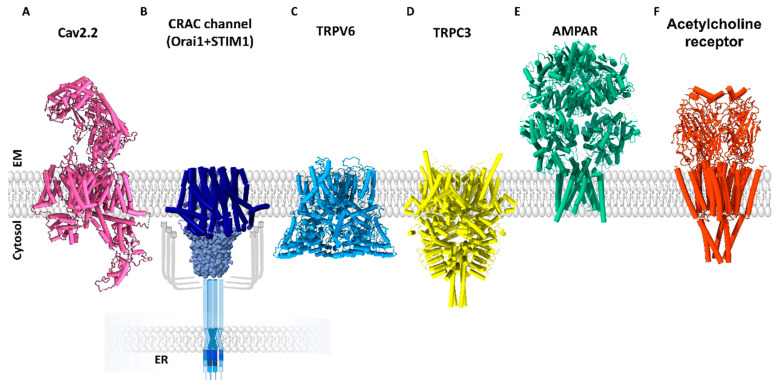
Structures of Ca^2+^ ion channels. Side views of the structures of the following ion channels, which are representative of the ion channel family to which they belong: (**A**) Voltage-gated Ca^2+^ ion channel: Cav2.2 (PDB:7MIY); (**B**) CRAC channel: Orai1 embedded in the plasma membrane and activated by direct coupling of STIM1 located in the endoplasmic reticulum; (**C**) Transient receptor potential vanilloid 6 (TRPV6) channel (PDB: 7K4A); (**D**) Transient receptor potential canonical 3 (TRPC3) channel (PDB: 5ZBG); (**E**) α-amino-3-hydroxy-5-methyl-4-isoxazolepropionic acid (AMPA) receptor (PDB: 5IDE); (**F**) Acetylcholine receptor (PDB: 2BG9).

**Figure 2 cells-10-03340-f002:**
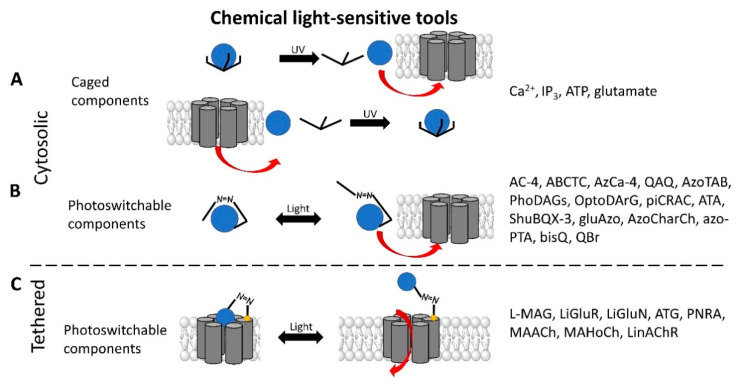
Schematic and examples for chemical light-sensitive tools: (**A**) Cytosolic caged components release a caged group or capture a Ca^2+^ ion upon irradiation with UV light. (**B**) Cytosolic and (**C**) tethered photoswitchable ligands which switch between two conformations upon irradiation with light of alternating wavelengths.

**Figure 3 cells-10-03340-f003:**
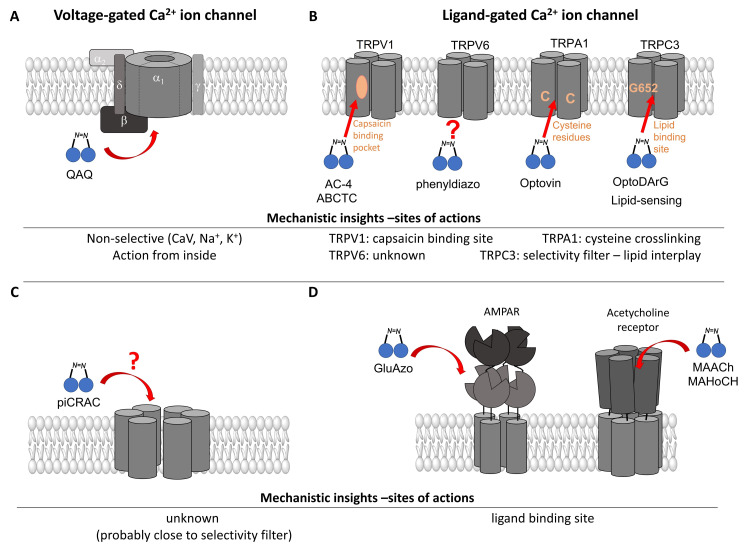
Scheme highlighting known or potential working mechanisms or sites of action of photoswitchable ligands: (**A**) The quaternary ammonium ion (QAQ) acts as an intracellular pore blocker of CaV channels. (**B**) Photoswitchable TRPV1 agonists/antagonists (AC-4, ABCTC) likely modulate ion channel function via binding the capsaicin-binding pocket. The photoswitchable rhodanine derivative, optovin, modulates TRPA1 function via covalent thioether formation with redox-sensitive key cysteines (Cys 621, 633 and 856) in the channel. OptoDArG, a highly efficient photochromic ligand for TRPC3, acts via a lipid sensing domain near the selectivity filter. Based on a TRPV6 inhibitor, a photoswitchable ligand appended with a phenyldiazo was shown to allow rapid photoswitching, however the site of action is still unknown. (**C**) piCRAC-1, based on the GSK-5498A compound, was shown to reversibly switch CRAC channels, however the site of action is unknown. (**D**) The soluble photoswitchable glutamate and acetylcholine receptor ligands acts via the corresponding ligand binding domain.

**Figure 4 cells-10-03340-f004:**
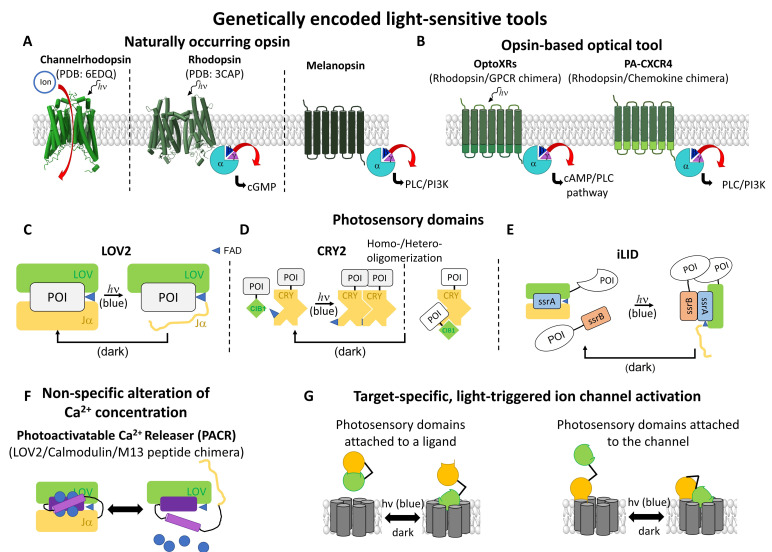
Spectrum of light-sensitive genetically encoded tools: (**A**) Naturally occurring opsins including channelrhodopsin (PDB: 6EDQ), rhodopsin (PDB: 3CAP), and melanopsin. (**B**) Based on opsins, e.g., OptoXRs and PA-CXCR4 were developed as optical tools to manipulate various cellular signaling cascades depending on cAMP/PLC or PLC/PI3K pathways. (**C–E**) Naturally occurring photosensory domains include the LOV2 domains (**C**), the CRY2 domains (**D**), and the iLID domain generated via the dimerizing peptides SsrA and SsrB (**E**). (**F**) Photoactivatable Ca^2+^ releaser (PACR) composed of a LOV2 domain, CaM, and a M13 peptide enables light-triggered release of Ca^2+^. (**G**) Possibilities for target-specific, light-triggered ion channel activation via photosensory domains, either only attached to an active site, or to both the ion channel and an active site. While the active site is hidden in the dark state, it is released upon blue light irradiation.

**Figure 5 cells-10-03340-f005:**
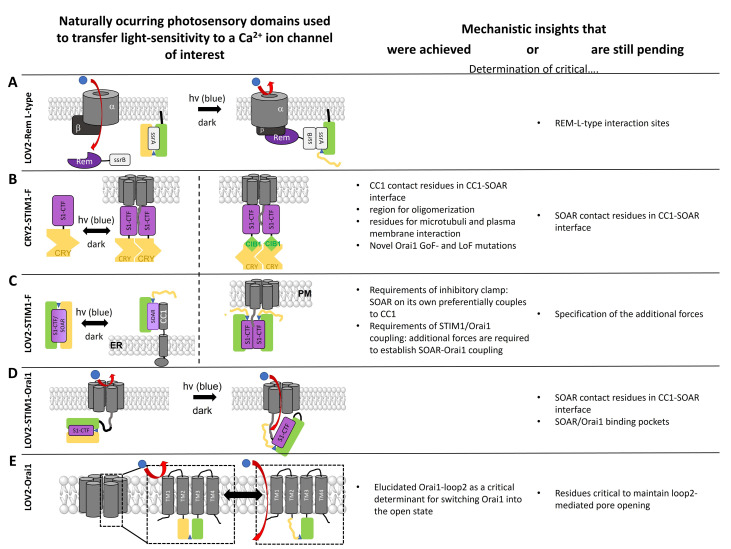
Schematic summarizing the published applications of photosensory domains to ion channels, the mechanistic insights that were achieved, as well as those which are still pending: (**A**) LOV2-RGK reversibly blocks L-type channels upon irradiation with UV light, however the site of action is still unknown. (**B**) CRY2-C-terminal STIM1-fragment (CRY2-STIM1-F/CRY2-CTF) or CRY2/CIB1-STIM1-F enable light-triggered homo- or heteromerization and thereby couple to Orai1 to trigger its activation. This approach enabled us to obtain detailed characterization of the CC1-SOAR contact interface, critical oligomerization sites, key determinants for oligomerization, and the identification of novel GoF- and LoF mutations. While the contacts for CC1 within the CC1-SOAR interface, those in the SOAR region still require further investigations. (**C**) LOV2-STIM1-F hides the active site of SOAR in the dark state, while, upon irradiation with UV light, it is released. Light-mediated release of SOAR also enables coupling to CC1 under resting cell conditions and coupling to Orai1 in store-depleted cells. LOV2-STIM1-F enabled to us determine the requirements for the inhibitory clamp and STIM1/Orai1 coupling. (**D**) Chimera of LOV2-STIM1-F and Orai1 enabling blue light triggered Orai1 activation. (**E**) Light-switchable Orai1 containing the LOV2 domain in the loop2 region. This suggests that the loop2 is a critical determinant for Orai1 pore opening, however critical residues remain to be examined.

**Table 1 cells-10-03340-t001:** Light sensitive chemical tools that modulate cytosolic Ca^2+^ levels. For each tool, the chemical structure, the target, as well as expressed cell line/organism are summarized.

Cages	Photosensitive Tool	Chemical Structure	Cell Line/ Organism	Ref.
Ca^2+^	Diazo-2	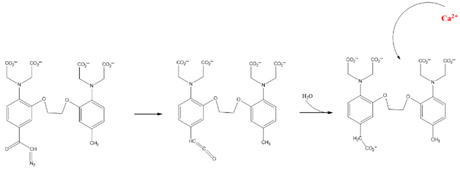	fibroblasts	[114]
	Diazo-4	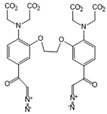	rat hippocampal pyramidal cells Aplysia neurons	[118,119]
	Nitr chelators	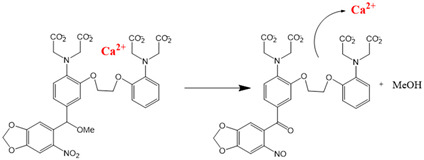	Cell bodies of cultured neurons of the fresh water snail Helisoma trivolvis	[125]
	DM-nitrophen	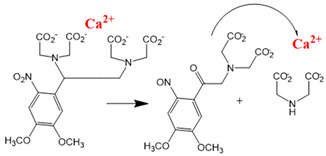	skinned rabbit psoas fibers	[126,127]
ATP	O-caged ATP-[γS]	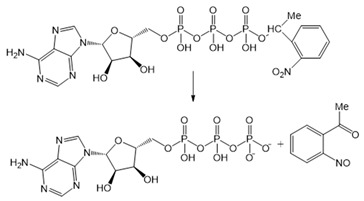	Hippocampal CA1 Pyramidal Neurons	[128]
glutamate	gluEPSPs	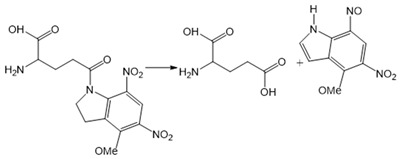	Hippocampal CA1 Pyramidal Neurons	[129]
IP_3_		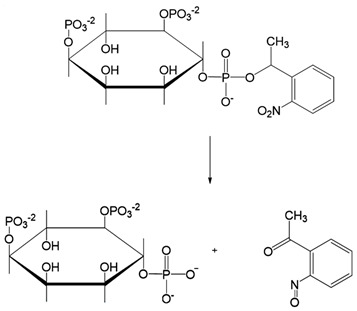	frog skeletal muscle	[130]

**Table 2 cells-10-03340-t002:** Light sensitive chemical tools that interfere with Ca^2+^ ion channels. For each tool the chemical structure, the target, the switching wavelength, the interfering mechanism, as well as expressed cell line/organism are summarized.

Photosensitive Tool	Chemical Structure	Target	wavelength	Mechanism	Cell Line/ Organism	Ref.
QAQ	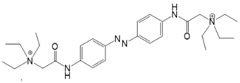	blocks voltage-gated K^+^, Na^+^ and Ca^2+^channels of TRPV1-expressing neurons	380 nm/ 500 nm light	light-dependent blocker of the Nav and Kv channel in its trans configuration and acts only at the inner side, but not at the outer side of the ion channel.	Rat hippocampal neuron, HEK-293T cells, mouse DRG neurons, spinal cord slices	[149]
AzoTAB	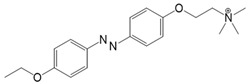	blocks voltage-gated Ca^2+^ ion channels	365 nm/ 490 nm	Its trans-isomer blocks NaV and CaV channels, resulting in the suppression of the spontaneous electric excitability in the cardiomyocytes.	Neonatal Rat Cardiomyocytes	[152]
AC-4	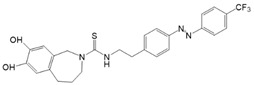	TRPV1	360 nm/ 440 nm	AC-4 is able to inhibit the TRPV1 channel. It acts as a trans-antagonist on the voltage-activated channel and as a cis-antagonist in experiments with the agonist capsaicin (TRPV1 agonist)	dorsal root ganglion neurons and C-fibre nociceptors, HEK293T cell	[153]
ABDTC	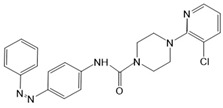	TRPV1	370 nm/ 470 nm	Cis-antagonist on voltage-activated TRPV1	HEK293T cells	[154]
optovin	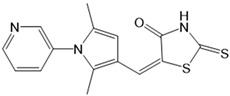	TRPA1	405 nm/ dark	Control TRPA1 activity by a light-activated intermediate state that reversibly forms covalent thioether with redox-sensitive key cysteines in the channel.	HEK293T cells, dorsal root ganglion sensory neurons, zebrafish, TRPA1-KO mice, DRG neurons, human cardiomyocytes	[155,156]
PhoDAGs	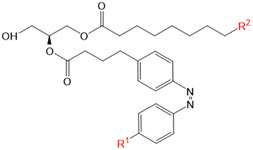	TRPC2/3/6	365 nm/ 470 nm	The photoswitchable diacylglycerols have photoswitchable fatty acids that enable the optical manipulation of TRPC2/3/6 activity.	HEK293 cells, mammalian tissue slices, mouse vomeronasal sensory neurons (VSNs), murine vomeronasal organ tissue slices	[157]
OptoDArG	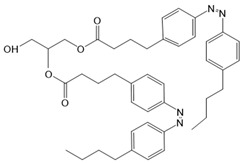	TRPC3	365 nm/ 430 nm	Photochromic ligand for TRPC3 channel that assists the activation of TRPC3 mutant, containing the G652A substitution near the selectivity filter.	HEK293 cells	[158]
piCRAC-1	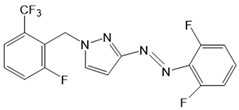	CRAC channel	365 nm/ 415 nm	The amide group of the CRAC channel inhibitors (e.g: GSK, Synta 66) is substituted by azo-group to convert it into photoswitchable derivatives.	HEK293 cells, zebrafish embryos	[159]
ATA-3	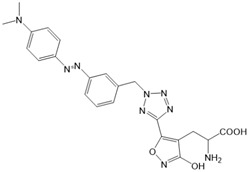	AMPA receptors	440 nm-480 nm/ dark	Its trans-isomer binds to the ligand-binding domains within AMPA receptors and allows for full blockage of the receptor, whereas the cis-isomer breaks up quickly.	HEK293T cells, mouse cortical slices, hippocampal neurons, TKO mouse retina	[160,161]
ShuBQX-3	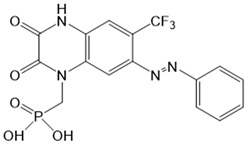	AMPA receptors	460 nm/ 600 nm	ShuBQX-3 is supposed to attach to the glutamate receptor ligand-binding domain	HEK293T cells, Xenopus oocytes, hippocampal neurons	[162]
gluAzo	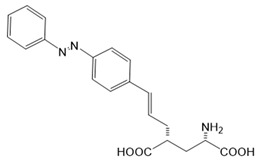	Kainate receptor	380 nm/ 500 nm	Because the binding pocket of the glutamate moiety of GluAzo photoswitch is analogous to that of soluble glutamate, it can activate and inhibit the kainate receptors reversibly.	rat hippocampal neurons, HEK293 cells, purkinje cells	[163,164,165]
LiGluR	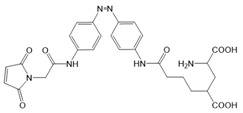	Kainate receptor	460 nm or 840 nm (two-photon)/ dark	It can induce the pore opening of the kainate receptor variants in an allosteric manner because it is located in close proximity to the ligand-binding domain	Bovine chromaffin cells, astrocytes, visual cortex of awake mice, zebrafish, hippocampal neurons, HEK293T cell	[166,167,168]
LimGluRs	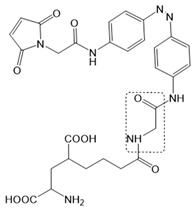	Kainate receptor	380 nm or 820 (two-photon)/ 500 nm	It can induce the pore opening of the kainate receptor variants in an allosteric manner because it is localized in close proximity to the ligand-binding domain	HEK293T cell, shippocampal neurons, zebrafish larvae, astrocytes, chromaffin cells, TKO mice	[169]
LiGluN	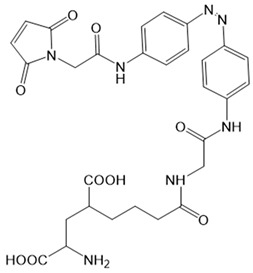	Activates or inhibit NMDA receptors (GluN1, GluN2A, GluN2B)	360 nm-405 nm (two-photon)/ 460 nm-560 nm	It can trigger the pore opening of the kainate receptor variants in an allosteric manner because it is localized in close proximity to the ligand-binding domain	HEK293T cells, hippocampal slices, larval zebrafish, GluN2A-knockout neonate mice	[170]
AzoCharCh	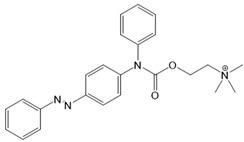	nAChRs	UV	AzoCharCh is a derivative of the carbamylcholine (potent depolarizing agent) that can reversibly inhibit acetylcholinesterase activity and function at low concentrations as receptor inhibitors block the depolarization generated by CharCh.	Electrophorous Electroplaques	[171]
Azo-PTA	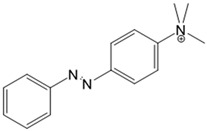	nAChRs	UV	Azo-PTA is a derivative of the phenyltrimethylammonium (potent depolarizing agent) that can reversibly inhibit acetylcholinesterase activity and function at low concentrations as receptor inhibitors block the depolarization generated by CharCh.	Electrophorous Electroplaques	[171]
bisQ	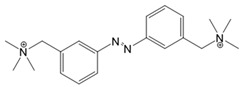	nAChRs	UV	Its trans-isomer is a potent activator of Electrophorus electroplaques, in which it can depolarize the electrogenic membrane of the electroplaque. It functions as inhibitor of acetylcholinesterase.	Electrophorous Electroplaques	[172]
QBr	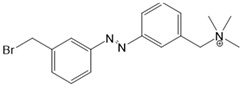	nAChRs	UV	trans-QBr can covalently bind to the electroplaques membrane and depolarize the membrane. It functions as an inhibitor of acetylcholinesterase.	Electrophorous Electroplaques	[172]
MAACh	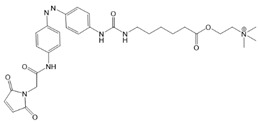	nAChRs	380 nm/ 500 nm or dark	The MAACh is used as a photoswitchable agonist for the nAChR receptor.	Xenopus oocytes	[173]
MAHoCh	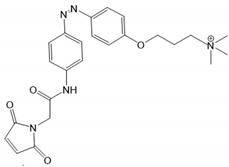	nAChRs	380 nm/ 500 nm or dark	The MAHoCh is a light-activated antagonist for the nAChR receptor.	Xenopus oocytes	[173]

## Data Availability

No new data were created or analyzed in this study. Data sharing is not applicable to this article.

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
