# Peer review of "Deciphering Molecular Mechanisms and Intervening in Physiological and Pathophysiological Processes of Ca2+ Signaling Mechanisms Using Optogenetic Tools"

_cells, 2021, doi:10.3390/cells10123340_

Round 1

Reviewer 1 Report

The manuscript is a rather detailed and complex review of old and novel tools and methods of optogenetics. Because of the complexity, the manuscript is very long, but I don't think the authors should shorten it. 
But this is the reason for a big number of formatting mistakes.
1. All tables contain have a problemы with hyphenation, spaces between words, and line spacing. Please fix it.
2. Legend of figure 5 is very far from the figure. It should be 2 lines closer. 
3. Subtitles of 4th order look very strange, as usual text. I think editors would not mind if you show them italic or bold,  or put before them letters (a. b. and etc).

Author Response

We thank the reviewer for the positive evaluation of our manuscript.

The manuscript is a rather detailed and complex review of old and novel tools and methods of optogenetics. Because of the complexity, the manuscript is very long, but I don't think the authors should shorten it.

But this is the reason for a big number of formatting mistakes.

  1. All tables contain have a problemы with hyphenation, spaces between words, and line spacing. Please fix it.

We used now landscape format for table 1 and table 2 and corrected hyphenation, spaces between words, line spacing.

  1. Legend of figure 5 is very far from the figure. It should be 2 lines closer.

This is adapted.

  1. Subtitles of 4th order look very strange, as usual text. I think editors would not mind if you show them italic or bold, or put before them letters (a. b. and etc).

We adapted according to reviewers comment.

Reviewer 2 Report

The manuscript "Deciphering molecular mechanisms and intervening in physiological and pathophysiological processes of Ca2+ signaling mechanisms using optogenetic tools" by Lena Maltan , Hadil Najjar , Adéla Tiffner and Isabella Derler is very interesting and well organized.    I have only one concern about paragraph 2.1.3 that should be better described. In particular, the mechanisms reported in the scheme should be better detailed in the text with particular attention to that of piCRAC.   

Author Response

We thank the reviewer for the positive evaluation of our manuscript.

The manuscript "Deciphering molecular mechanisms and intervening in physiological and pathophysiological processes of Ca2+ signaling mechanisms using optogenetic tools" by Lena Maltan , Hadil Najjar , Adéla Tiffner and Isabella Derler is very interesting and well organized.    I have only one concern about paragraph 2.1.3 that should be better described. In particular, the mechanisms reported in the scheme should be better detailed in the text with particular attention to that of piCRAC.   

We adapted the paragraph 2.1.3, improved that connection between the text and Figure 3 and provided a more detailed explanation of the potential mechanism of piCRAC.

Reviewer 3 Report

This review on the optogenetic tools to understand the molecular mechanisms regulating the Ca2+ signalling involved in physiological and pathophysiological processes is rather complete and well written. The Authors first briefly describe the Ca2+ entry pathways (VGCCs, store-dependent Ca2+ ion channels, TRP channels and receptor Ca2+ ion channels) and then describes in details the optogenetic tools developed to regulate Ca2+ signals. I appreciated the description of the different opsin  used on Ca2+ signaling, particularly of channelrhodopsin-2.

The chapter on the physiological and pathophysiological relevance of novel optical tools in the field of 858 Ca2+ permeable ion channels, both in vitro and in vivo conclude nicely the presentation of the importance of the optogenetic tools for potential thearapies.

I do not have main criticisms. I would like just to point out that in the first sentence of the Abstract are missing several functions controlled by Ca2+ channels: neurotrasmission, excitation, hormone secretion, memory, aging....

While describing the VGCCs Ca2+ channel subunits regulate also N-type channels;. in general presynaptic Ca2+ channels (see the papers on the role of alpha2-delta subunit).

Author Response

We thank the reviewer for the positive evaluation of our manuscript.

This review on the optogenetic tools to understand the molecular mechanisms regulating the Ca2+ signalling involved in physiological and pathophysiological processes is rather complete and well written. The Authors first briefly describe the Ca2+ entry pathways (VGCCs, store-dependent Ca2+ ion channels, TRP channels and receptor Ca2+ ion channels) and then describes in details the optogenetic tools developed to regulate Ca2+ signals. I appreciated the description of the different opsin  used on Ca2+ signaling, particularly of channelrhodopsin-2.

The chapter on the physiological and pathophysiological relevance of novel optical tools in the field of 858 Ca2+ permeable ion channels, both in vitro and in vivo conclude nicely the presentation of the importance of the optogenetic tools for potential thearapies.

I do not have main criticisms. I would like just to point out that in the first sentence of the Abstract are missing several functions controlled by Ca2+ channels: neurotrasmission, excitation, hormone secretion, memory, aging....

As suggested, we extended the abstract by the additionally known functions of Ca2+ ion channels.

While describing the VGCCs Ca2+ channel subunits regulate also N-type channels;. in general presynaptic Ca2+ channels (see the papers on the role of alpha2-delta subunit).

We adapted that among Ca2+ ion channels not only L-type, but also the other types, in particular R-, P/Q- and N-type channels, can be regulated by auxiliary subunits.